# Reactivation of a somatic errantivirus and germline invasion in *Drosophila* ovaries

Marianne Yoth[1], Stéphanie Maupetit-Méhouas[1], Abdou Akkouche[1], Nathalie Gueguen[1], Benjamin Bertin[2], Silke Jensen [1] ✉ & Emilie Brasset [1] ✉

Most *Drosophila* transposable elements are LTR retrotransposons, some of which belong to the genus Errantivirus and share structural and functional characteristics with vertebrate endogenous retroviruses. Like endogenous retroviruses, it is unclear whether errantiviruses retain some infectivity and transposition capacity. We created conditions where control of the *Drosophila ZAM* errantivirus through the piRNA pathway was abolished leading to its de novo reactivation in somatic gonadal cells. After reactivation, *ZAM* invaded the oocytes and severe fertility defects were observed. While ZAM expression persists in the somatic gonadal cells, the germline then set up its own adaptive genomic immune response by producing piRNAs against the constantly invading errantivirus, restricting invasion. Our results suggest that although errantiviruses are continuously repressed by the piRNA pathway, they may retain their ability to infect the germline and transpose, thus allowing them to efficiently invade the germline if they are expressed.

Long Terminal Repeat (LTR)-retroelements are a class of transposable elements (TEs) that inhabit nearly all eukaryotic genomes. The genome of *Drosophila* is largely occupied by a group of insect endogenous LTR-retroelements called errantiviruses, which belong to the Metaviridae family. Errantiviruses share structural and functional characteristics with endogenous retroviruses (ERVs), which are vertebrate LTR-retroelements. Most errantiviruses encode 3 open reading frames (ORFs) whose functions are analogous to that of the Gag, Pol and Env proteins of ERVs or retroviruses. ERVs derive from exogenous retroviruses that integrated into the host germline genome, became permanent elements and are then vertically transmitted. Therefore, they represent a partial record of previous retroviral infections; in Human, ERV-related sequences occupy 9% of the genome, in mice 12%, in rat 10.2%, in microbats 5.3%, in marmosets 7.5% (http://repeatmasker.org/genomicDatasets/RMGenomicDatasets.html). Following integration, ERVs are then capable of replicating themselves and integrating at new genomic loci. Each new ERV copy evolves independently and can be at the origin of the emergence of a new TE family. In mice, IAPE and IAP ERVs illustrate the evolutionary history of ERVs[1]. IAPEs are retroviruses capable of infecting other cells after an extracellular passage, whereas highly-repeated IAPs, derived from IAPEs by losing the envelope gene

(*env*), are strictly intracellular. Notably, the majority of ERVs are incapable of producing infectious viral particles in vivo. However, insect errantiviruses are not thought to have infectious retroviruses as ancestors and have rather gained potential infectivity independently through the acquisition of a Baculovirus *env* gene[2].

It is crucial to note that TEs can only be maintained in a species if they are able to transpose into the germline genome. If they do not, they will accumulate mutations until there are no functional copies left, and the TE family dies out. However, some TEs, including ERVs and errantiviruses, gained cell and developmental stage specific expression[3–7]. For instance, in the *Drosophila melanogaster* genome, *ZAM*, *Idefix* or *Gypsy* errantiviruses are expressed exclusively in the somatic follicle cells of the ovaries, with somatic transcription factors controlling their expression[8–15]. The *Drosophila melanogaster* ovary is comprised of about 16 ovarioles, each of which contains a succession of follicles composed of germ cells surrounded by somatic follicle cells[16]. To reach the germline genome, these errantiviruses must then cross the so-called Weismann barrier separating somatic and germ cells. *Gypsy* and *ZAM* virus-like particles have been observed in somatic follicle cells. They likely contain the genomic RNA and the expression of the envelope protein could mediate an infection process[8,17].

[1]iGReD, Université Clermont Auvergne, CNRS, INSERM, Faculté de Médecine, 63000 Clermont-Ferrand, France. [2]LIMAGRAIN EUROPE, Centre de recherche, 63720 Chappes, France. ✉e-mail: silke.jensen@uca.fr; emilie.brasset@uca.fr

However, the *Gypsy* envelope is not involved in the soma-to-germline transfer[15]. Instead, it has been proposed that not only *Gypsy* but also *ZAM* hijack the host vitellogenic pathway to target the oocyte[11,18].

Although the presence of TEs in the genome has been shown to provide some evolutionary benefits (reviewed in[19]), unregulated TE expression and transposition represent a threat to genome integrity and host fitness. In metazoan gonads, TE expression and mobilization is restrained by PIWI-interacting RNAs (piRNAs) (reviewed in[20,21]). piRNAs originate from specific source loci, the piRNA clusters[22]. In *Drosophila* gonads, the piRNA clusters expressed in the germline are particularly numerous and diverse, while there is only one major piRNA cluster expressed in gonadal somatic cells, that is *flamenco* (*flam*)[23–27]. Research has primarily concentrated on how piRNAs silence TEs that are expressed in the cells where the corresponding piRNAs are produced. The role of piRNAs in the silencing of a TE that is activated in a cell type different from the one where the silencing mechanism must take place remains unclear. Consequently, whether these TEs can escape the robust epigenetic control mediated by piRNAs when arriving in the germline and how they are tamed by the host over time remained to be puzzled out.

The *ZAM* errantivirus was discovered through its uncontrolled activity in an unstable line, RevI-H2, about 30 years ago[28,29]. *ZAM* has 3 intact ORFs Gag, Pol and Env[28]. In the RevI-H2 *Drosophila* line bearing a large deletion in the *flamenco* piRNA cluster, *ZAM* had become active and is specifically expressed in gonadal somatic cells[8,25]. Moreover, *ZAM* inserted into multiple new loci in the RevI-H2 line, including a germline-specific piRNA cluster, resulting in unexpected production of germline piRNAs that map to *ZAM*[28–30]. The biological role of these germline *ZAM*-mapping piRNAs remained unexplored.

Since these studies, *ZAM* has emerged as a key model for studying host-TE interactions and the relationship between somatic cells and the germline during TE invasion. By analyzing different conditions of *ZAM* reactivation, we aimed here to study what happens when an errantivirus is reactivated de novo and how the host responds. Specifically, we sought to elucidate whether piRNAs produced in the germline counteract TE invasion from the somatic cells or whether TE transcripts are protected from degradation by virus-like particles when arriving in the germline.

We show that in the RevI-H2 line, *ZAM* expression persists in the follicle cells but ZAM does not invade the germline, while *ZAM* copies with invasive capacities have been maintained in the genome. To recreate the initial unstable condition, we de novo reactivated *ZAM* in the ovarian follicle cells, either by soma-specific knock-down of the piRNA pathway, or by deleting a *ZAM* copy in the somatic *flamenco* piRNA cluster, while keeping the piRNA pathway fully functional. De novo *ZAM* reactivation in the follicle cells led to a massive *ZAM* invasion deep into the adjacent oocyte and its ooplasm. We demonstrate that this invasion could be impeded by the expression of de novo *ZAM*-targeting piRNAs produced in the germ cells themselves, demonstrating that these piRNAs are functional against the native *ZAM*. These results show that by being expressed exclusively in somatic cells, errantiviruses may evade the highly efficient control of the germline piRNA pathway and remain active for long periods of time (in the case of the RevI-H2 line, at least 30 years). The challenged germ cells then mount an adaptive genomic immune response to tackle the permanent invasion.

## Results

### In an ancient *flamenco* mutant line, ZAM is still expressed, but the line is nevertheless stable

To investigate the history of *ZAM* transposition dynamics, we monitored *ZAM* copies in the RevI-H2i2 line, an isogenic line that was recently derived from the parental *flam* mutant RevI-H2 line that has more than 25 years of laboratory history. It had previously been reported that the RevI-H2 line was unstable and that *ZAM* actively

transposed in this line[29]. The *ZAM* instability in RevI-H2 had been linked to a deletion of a large part of the *flam* piRNA cluster. This deletion spanned over more than 120 kb and comprised many different TE relics, including more or less recent copies, and all *ZAM*-related copies of the *flam* locus[25,26].

In the RevI-H2i2 genome, Oxford Nanopore Technology (ONT) genome sequencing revealed 18 *ZAM* copies (Fig. 1a and Supplementary Data 1 for coordinates). Only one of these *ZAM* copies was also present in the reference genome (http://flybase.org). The other 17 new insertions were very similar to the *ZAM* reference element identified in the initial RevI-H2 line[28] and we could clearly localize 14 of the 17 new *ZAM* insertions. Interestingly, ONT long-read sequencing, allowed identifying different *ZAM* variants. Specifically, six were full-length *ZAM* elements (ZAM-fl), one, named ZAM-v1, harbored a deleted 5'-UTR (5' untranslated region), two, named ZAM-v3, had a 303 bp deletion within the 5'-UTR, four had large internal deletions in the coding regions, and four were full-length insertions with a deletion at the C-terminal end of the *pol* gene of positions 5494 to 6120 in the *ZAM* internal sequence[28] (Repbase ZAM_I, https://www.girinst.org/repbase/[31]). This deleted part of the *pol* gene does not correspond to any known protein domain, and the fact that there are five identical copies of this *ZAM* at different locations in the RevI-H2i2 genome indicates that this *ZAM* variant, which we named ZAM-v2, is competent for transposition. The full-length *ZAM*, ZAM-v1 and ZAM-v3 copies all potentially encode Gag, Pol and Env proteins (Fig. 1a).

In the initial RevI-H2 line, a *ZAM* insertion was found in the germline piRNA cluster *9*, which is close to the X chromosome centromere[30]. Here, in the RevI-H2i2 line, thanks to long-read sequencing, we could identify *ZAM* insertions in repeated sequences, such as piRNA clusters, more accurately. We now identified three *ZAM* insertions in piRNA cluster *9* (one ZAM-fl and two ZAM-v2), and also a ZAM-fl insertion, which is localized in either piRNA cluster *13* or cluster *56* (uncertainty is due to flanking repeats, piRNA clusters as in[32]) (Fig. 1a). We reanalyzed the Illumina sequencing data of the initial RevI-H2 line and found evidence that all these *ZAM* insertions in piRNA cluster *9* and *13* or *56* were already present ten years ago, and that at least 14 of the *ZAM* insertions in the RevI-H2i2 line were already present in the parental RevI-H2 line. This result indicates that the genomic *ZAM* profile is the same as ten years ago and suggests that no new *ZAM* insertions occurred since then in the germline.

Strikingly, no new *ZAM* insertion was detected in the *flam* piRNA cluster. Thus, in the RevI-H2i2 line, no *ZAM* piRNA can be produced from *flam*, the major somatic piRNA source locus. Interestingly, single-molecule RNA fluorescence in situ hybridization (smRNA FISH) revealed the presence of *ZAM* RNA in RevI-H2i2 follicle cells, demonstrating that *ZAM* is not silenced in these somatic cells and that transcriptionally active copies of *ZAM* are still present in the genome. *ZAM* transcripts were produced in follicle cells, starting very early during oogenesis in the germarium. Then, after stage 8, *ZAM* transcripts accumulated in a patch of follicle cells located at the posterior side of the follicles. We did not detect any *ZAM* staining in germ cells (nurse cells or oocytes) (Fig. 1b). RNase A treatment led to complete loss of *ZAM* staining in the follicle cells (Supplementary Figure 1a). We did not detect any *ZAM* RNA in the Iso1A, $w^{1118}$ and $w^{IR6}$ control ovaries (Supplementary Figure 1b). Using RT-qPCR, we confirmed that *ZAM* was expressed in the RevI-H2i2 ovaries. Conversely, other TEs such as the soma-specific *Gypsy* and the germline-specific *Burdock*, were not upregulated (Fig. 1c). Interestingly, we observed that, besides the full-length ZAM, at least *ZAM* variants ZAM-v2 and ZAM-v3 are expressed in RevI-H2i2 ovaries (Supplementary Figure 1c, d). *ZAM* is an errantivirus that encodes Gag, Pol and Env proteins. Using immunostaining, we were able to detect *ZAM* Gag and Env proteins that accumulated in the posterior follicle cells in late-stage follicles ( > stage 9) (Fig. 1d). All these results showed that *ZAM* transcripts and proteins are still expressed in the RevI-H2i2 line. Although *ZAM* was expressed in follicle

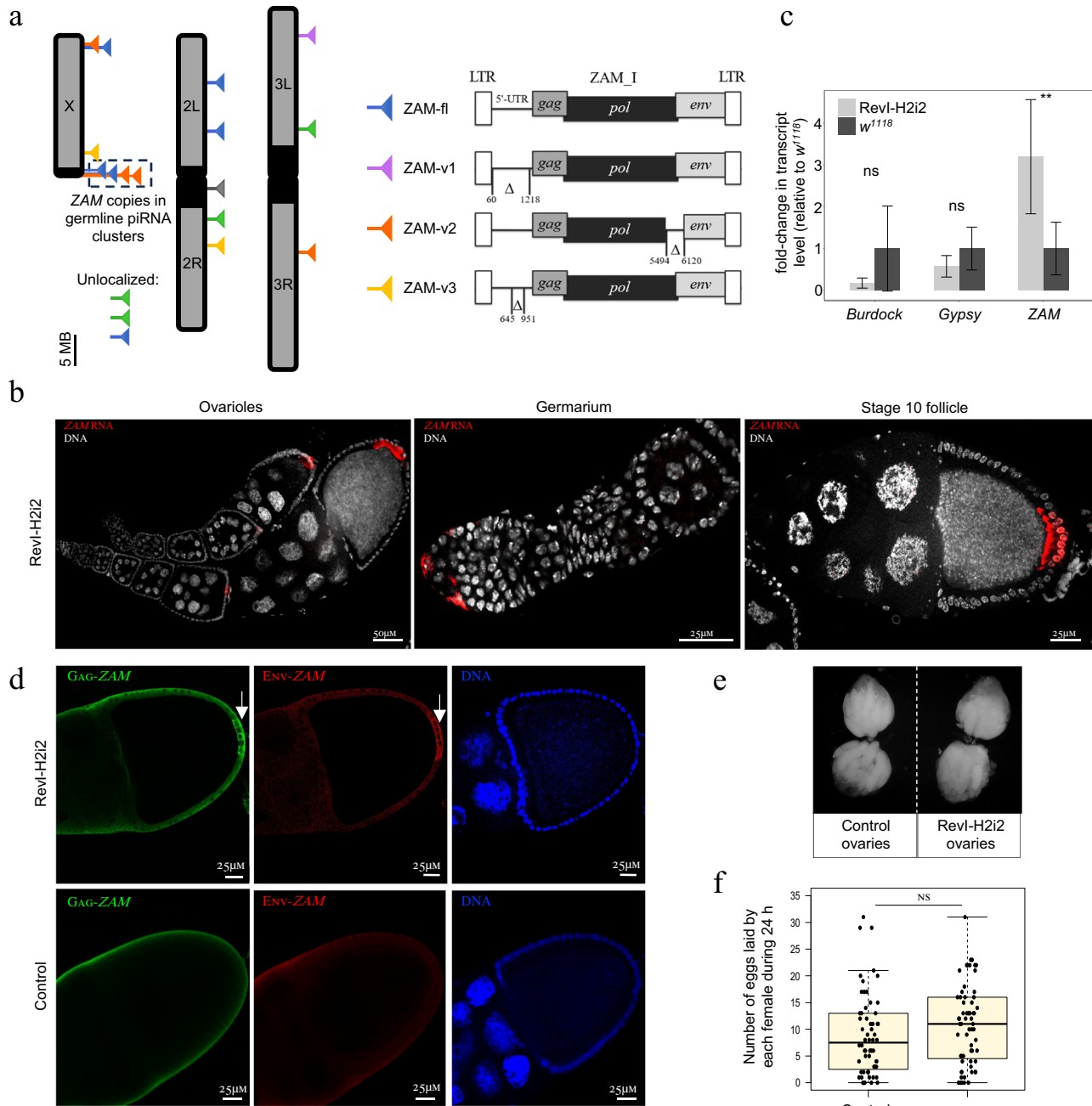

**Fig. 1 | *ZAM* is expressed in follicle cells in the RevI-H2i2 isogenic line but the line is stable. a** *ZAM* elements in the RevI-H2i2 genome. Each triangle represents a *ZAM* insertion. The different *ZAM* variants are presented in the illustration on the right. The positions refer to *ZAM* internal sequences (ZAM_I in RepBase). Blue: full-length *ZAM* elements (ZAM-fl); violet: *ZAM* variants ZAM-v1 with deleted 5'-UTR; orange: *ZAM* variants ZAM-v2, in which positions 5494-6120 are deleted; yellow: ZAM variants ZAM-v3 with a 303 bp deletion in the 5'-UTR; green: *ZAM* with internal deletions other than *ZAM* variants above; gray: full-length *ZAM* copy also present in the reference Release 6 genome (http://flybase.org). The dotted box indicates four *ZAM* copies on the X chromosome inserted in referenced germline piRNA clusters (Cluster *9* and *13*). No *ZAM* insertion was found in chromosome 4 (1.35 Mb, see Supplementary Data 1). For coordinates and details see Supplementary Data 1. **b** Projection of confocal images of ovarioles, germarium, and stage 10 follicles showing *ZAM* expression by smRNA FISH (in red) in RevI-H2i2 ovaries. DNA was stained with DAPI (white). The experiment was independently repeated at least 10 times, with similar results obtained each time. **c** Fold-change in the steady-state *ZAM*, *Gypsy* and *Burdock* RNA levels for RevI-H2i2 ovaries compared with $w^{1118}$

ovaries (control), quantified by RT-qPCR (primer sequences in Supplementary Data 5). $n = 3$ biological independent samples and two reverse transcription experiments have been performed for each replicate, resulting in a total of six data points per genotype. All data points are shown as dot plots. **$p$ value = 0,0087; ns, not significant ($p$ value > 0.05) (Mann-Whitney two-sided test). Data are presented as mean values and the error bars indicate the standard deviation from the mean values. **d** Confocal sections of stage 10 egg chambers of RevI-H2i2 and $w^{1118}$ control line showing *ZAM*-encoded Gag (green) and Env (red) proteins. DNA was stained with DAPI (white). The experiment was independently repeated at least 5 times, with similar results obtained each time. **e** Morphology of control and RevI-H2i2 ovaries. **f** Box plot displaying the number of eggs laid per fly per day by control and RevI-H2i2 females. Each dot represents an individual female. The experiment was repeated twice for each condition to reach $n = 60$ control females, $n = 63$ RevI-H2i2 females. In the box plots, the midline corresponds to the median value; the lower and upper hinges correspond to the first and third quartiles; and the whiskers span the minimum and maximum values, excluding outliers. ns, not significant ($p$ value > 0.05) (Mann-Whitney two-sided test).

cells, the RevI-H2i2 line was fertile. Indeed, the ovary morphology and the number of eggs laid were comparable in RevI-H2i2 and control females (Fig. 1e, f). Altogether, these results indicate that no silencing mechanism had been set up to repress the *ZAM* errantivirus in follicle cells, suggesting that *ZAM* expression has no major deleterious effect in the RevI-H2i2 line and that there is no selective pressure to specifically repress *ZAM* in the follicles cells. These findings suggest that RevI-H2i2 is a stable line although *ZAM* is actively expressed in follicle cells.

### Germline *ZAM*-targeting piRNAs constrain *ZAM* invasion from adjacent somatic cells

Transposition of *ZAM* in the initial RevI-H2 line had occurred in the germline, as attested by the transmission of *ZAM* insertions to the offspring, despite the fact that *ZAM* is specifically and exclusively expressed in the somatic follicle cells. This confirms that the *ZAM* errantivirus, when expressed in follicle cells, transits to the germline to integrate into the germ cell genome. However, we showed that *ZAM* insertions are stabilized in the RevI-H2i2 line. The RevI-H2i2 flies produce high amounts of sense and antisense *ZAM*-derived piRNAs with a ping-pong signature, revealed by an over-representation of 10-nucleotide 5'-overlaps between sense and antisense *ZAM*-derived piRNAs (Fig. 2a), while there was no ping-pong signature for ZAM in diverse control lines[14,30]. Ping-pong amplification of piRNAs can only occur in the germline in *Drosophila* (reviewed in[33]). Thus, these *ZAM* piRNAs observed in RevI-H2i2 are derived from the germline. Importantly, our previous research has demonstrated that the X chromosome of the RevI-H2 line, which contains the *ZAM* copies inserted in germline piRNA clusters, is both necessary and sufficient to produce *ZAM*-regulating piRNAs[30]. Additionally, in both RevI-H2[30] and RevI-H2i2 background, a *ZAM* sensor transgene is silenced in the germ cells (Supplementary Figure 2a). We therefore hypothesized that piRNAs produced in germ cells can thwart *ZAM* invasion from somatic cells to the germline and thus protect the germline from new *ZAM* transposition. However, piRNAs are known to silence TEs in the cells where they are produced and some TEs, such as *ZAM*, are not expressed directly in germ cells but arrive from surrounding somatic cells. Moreover, *ZAM* RNA may transit in an encapsulated form and the capacity of piRNAs to target encapsulated RNAs remains unknown.

To test whether germline *ZAM*-derived piRNAs were efficient to counteract an invasion coming from surrounding somatic cells, we abolished the piRNA pathway in the germ cells by germline-specific knock-down (GLKD) of piRNA pathway components, in a RevI-H2i2 genetic background producing *ZAM* piRNAs in the germline. It is important to note that, during the genetic crosses performed, only the X chromosome of the RevI-H2i2 was tracked to ensure that the ZAM insertions in the germline piRNA clusters were maintained (*Drosophila* lines for knock-down are listed in Supplementary Data 2, crossing schemes presented in Supplementary Figure 6). We observed that the germline knock-down of proteins of the piRNA pathway, Zucchini (Zuc), Argonaute 3 (Ago3) and Piwi, led to a decrease in the production of *ZAM*-targeting piRNAs, i.e. piRNAs that are complementary and thus antisense to ZAM (Fig. 2b, Supplementary Figure 2b). The decrease in antisense piRNAs was comparable to that observed for germline TEs (i.e., *Burdock* and *Accord*) or intermediate TEs (i.e., *Idefix*), while the level of piRNA of somatic TEs, such as *Tirant*, was not affected. Furthermore, in *ago3*-GLKD ovaries, the ping-pong signature for *ZAM* was abolished, as for many other germline-specific TEs, while it was maintained in *piwi*-GLKD ovaries, for instance (Fig. 2c, Supplementary Figure 2c).

We then analyzed the subcellular localization of *ZAM* RNAs by smRNA FISH in the RevI-H2i2 *white*-GLKD control line in which *ZAM*-derived piRNAs are produced in the germline. *ZAM* RNA staining was restricted to follicle cells, only a minor *ZAM* RNA signal was detected in the posterior pole of approximately 10% of stage 10 oocytes (Fig. 2d, e). However, when the piRNA pathway was abolished in RevI-H2i2 germ

cells through *vreteno (vret)*, *zuc*, *ago3* or *armitage (armi)* GLKD, *ZAM* RNA was no longer restricted to follicle cells, a high signal was also detected in oocytes. *ZAM* RNA spread throughout the ooplasm, but was clearly enriched at the posterior pole of the oocyte, adjacent to the *ZAM*-expressing follicle cells, supporting the fact that *ZAM* RNA originated from follicle cells (Fig. 2d). Actually, 30-70% of stage 10 follicles showed a strong *ZAM* RNA signal in oocytes (Fig. 2e). The strongest invasion phenotype was observed for the RevI-H2i2 *ago3*-GLKD condition where we found a decrease in the production of *ZAM*-derived piRNAs and the loss of the ping-pong signature (Fig. 2b). In this condition, *ZAM* RNA accumulation in the ooplasm was correlated with a strong increase of *ZAM* RNA in total ovaries (RT-qPCR data) (Supplementary Figure 2d). These results showed that when the piRNA pathway is affected in the germline, *ZAM* RNAs transit from the somatic follicle cells, where they are produced, to the oocyte. Altogether, our data strongly suggest that *ZAM* piRNAs produced by the germline piRNA pathway trigger post-transcriptional silencing of *ZAM* RNAs arriving from follicle cells.

We confirmed this finding using a line where no *ZAM*-derived piRNA is produced in the germline (*w^1118^* genetic background) and in which we knocked down the piRNA pathway in the follicle cells by somatic knock-down (sKD) of *armi*, *vret*, *piwi* or *yb*. The amount of antisense piRNAs that target soma-specific TEs, including *ZAM*, was strongly decreased (Fig. 2f, Supplementary Figure 2e, f). RT-qPCR analysis showed that *ZAM* and another soma-specific TE, *Gypsy*, were derepressed, but not the germline-specific *Burdock* (Supplementary Figure 2g). Importantly, smRNA FISH results revealed that *ZAM* RNAs were not restricted to the posterior follicle cells. Like in ovaries harboring GLKD of the piRNA pathway in RevI-H2i2 background, *ZAM* RNAs were present in oocytes (Fig. 2g, h, Supplementary Figure 2h). We detected *ZAM* RNAs in the ooplasm in 90% of stage 10 follicles (Fig. 2i). This result confirmed that, when no *ZAM* piRNA is produced in the germline, *ZAM* RNAs expressed in the somatic follicle cells transit to the oocyte and then persist in the oocyte.

### *ZAM* RNAs transcribed in follicle cells transit to the oocyte and are conveyed to the embryos in the absence of germline ZAM piRNAs

Although *ZAM* is expected to be transcribed only in follicle cells due to its dependence on the Pointed somatic transcription factor[8,34], it was possible that some ZAM copies acquired the capacity to be expressed in the germline over time. Therefore, we wanted to rule out the possibility that *ZAM* RNAs detected in the RevI-H2i2 oocyte originated from germinal nurse cells. If some *ZAM* genomic insertions could be expressed in the germline of the RevI-H2i2 line, then germline depletion of *Piwi*, which is required for the transcriptional gene silencing of TEs, should lead to the transcriptional de-silencing of *ZAM*. We observed that Piwi-GLKD in the RevI-H2i2 ovaries led to the de-silencing of TEs that are capable of transcribing in the germline, such as *Burdock*. However, we did not observe any changes in either the *ZAM* RNA level or localization (Fig. 3a, b). Moreover, the depletion of *Ago3* induced a substantial accumulation of *ZAM* RNA solely in the ooplasm. In contrast to TEs expressed in the germline (e.g, *Burdock*), we never detected *ZAM* RNA staining in the cytoplasm of the nurse cells at any stage (Fig. 3c). These results strongly support that *ZAM* cannot be expressed in the RevI-H2i2 germ cells and that *ZAM* RNAs detected in oocytes originate from the somatic follicle cells, not nurse cells.

Throughout the animal kingdom, the first embryonic development stages are controlled by transcripts and proteins deposited by the mother during oogenesis[35]. In *Drosophila*, most of the maternal mRNAs are dumped from nurse cells into the oocyte during oogenesis. Although *ZAM* transcripts originate from the somatic follicle cells, when we investigated *ZAM* RNA transmission to the oocyte, we found such transcripts also in early embryos. Indeed, in conditions where

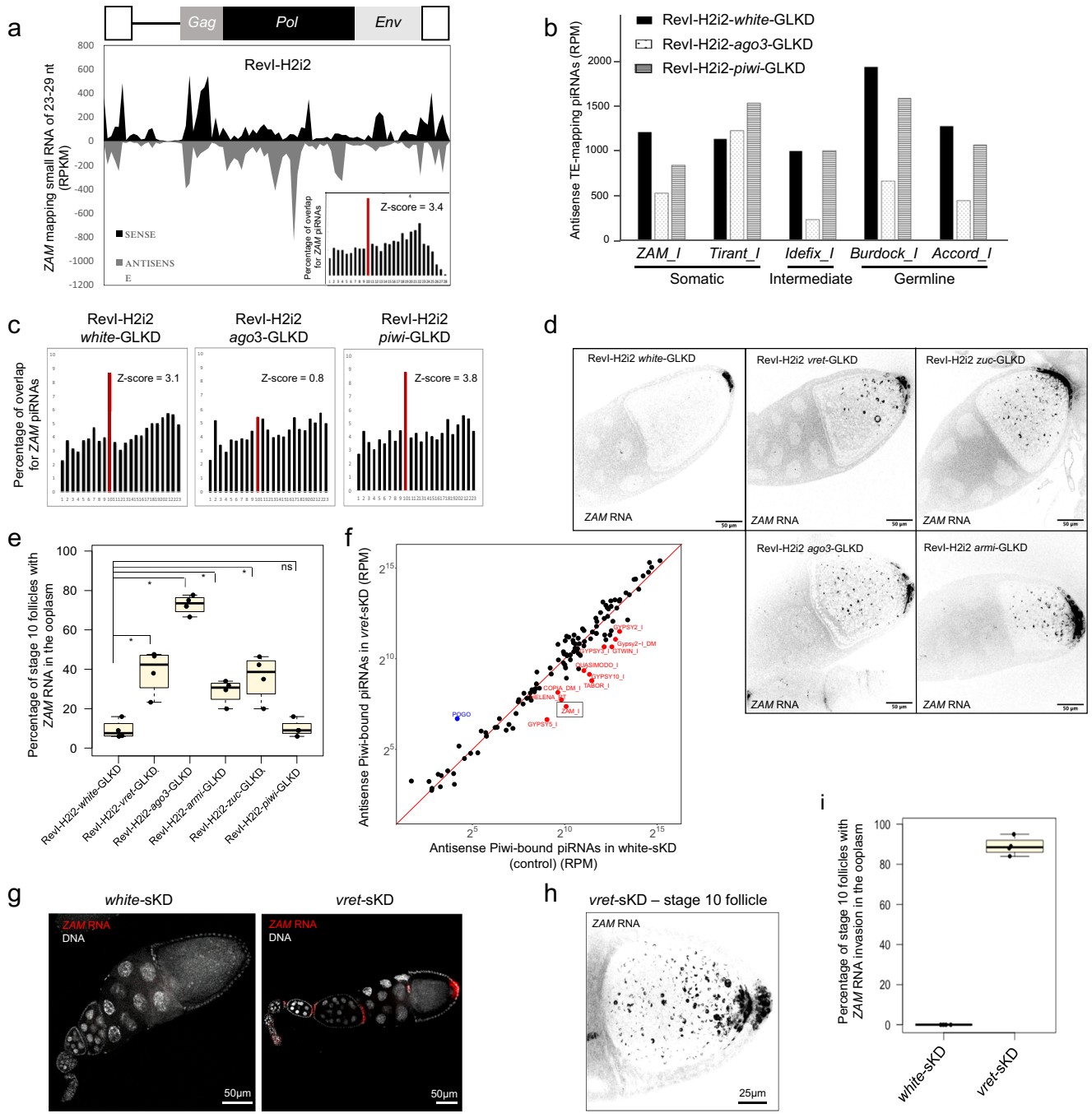

*ZAM* RNA had been detected in oocytes, smRNA FISH experiments revealed a strong accumulation of *ZAM* RNAs in early embryos, before the zygotic transition, 0-2 hours after egg laying: from a RevI-H2i2 mother with GLKD of the piRNA pathway, and also from a mother without germinal *ZAM* piRNAs with sKD of the piRNA pathway in follicle cells (Fig. 3d, Supplementary Figure 3a). Furthermore, *ZAM* RNAs accumulated at the posterior pole of early embryos where future germ cells cellularize (Fig. 3e, Supplementary Figure 3b). We detected *ZAM* RNA in high quantity until stage 5 of embryogenesis and few *ZAM* RNA signal persisting until the cellularization of the blastoderm ( ~ stage 8-9) (Fig. 3f). In the progeny of the RevI-H2i2 line, we did not detect any *ZAM* RNA, although *ZAM* RNAs were produced in follicle cells of RevI-H2i2 ovaries (Fig. 3d). This confirmed that *ZAM* invasion of the germ-line did not occur in this line. These data revealed an efficient post-transcriptional silencing of *ZAM* RNAs arriving from follicle cells into the oocyte by piRNAs produced in the germline. This mechanism

should limit the transposition of the somatic *ZAM* retrotransposon into the germline genome.

## De novo *ZAM* reactivation leads to massive oocyte invasion and may result in severe fertility defects

Our results demonstrate that an adaptive response can emerge to counteract the invasion of germ cells by an errantivirus. To go deeper, we sought to assess the direct impact of reactivating an errantivirus de novo, before the establishment of any adaptive response by the germline. To conduct this study, we aimed to specifically reactivate *ZAM* de novo. The loss of the piRNA pathway in the somatic follicle cells leads to *ZAM* reactivation, however, many other TEs are also desilenced (Supplementary Figure 2e, g). Moreover, alteration in the expression of piRNA pathway genes also results in severe developmental defects during oogenesis[36–39]. Therefore, this model cannot be used to specifically analyze the impact of *ZAM* reactivation in ovaries.

**Fig. 2 | Germline *ZAM* piRNAs produced in RevI-H2i2 germ cells counteract *ZAM* invasion. a** Density plot of *ZAM*-mapping regulatory piRNAs along the *ZAM* sequence in RevI-H2i2 ovaries (up to 3 mismatches). In the lower right corner is the histogram showing the percentage of 5'-overlaps between sense and antisense *ZAM*-derived piRNAs (23–29 nt) in RevI-H2i2 ovaries. The proportion of 10nt 5'-overlaps is in red and the corresponding Z-score is indicated. **b** Antisense piRNAs mapping to TE internal sequences (0-3 mismatches) in the RevI-H2i2 line upon *white*- (control), *ago3*-, or *piwi*-GLKD. Normalized per million of genome-mapping piRNAs. **c** Histogram showing the percentage of 5'-overlaps between sense and antisense *ZAM*-mapping piRNAs (up to 3 mismatches) in *white*-, *ago3*-, or *piwi*-GLKD RevI-H2i2 ovaries. The percentage of 10nt overlaps is in red and the corresponding Z-score is indicated. **d** Color-inverted confocal images of stage 10 egg chambers showing *ZAM* smRNA FISH signal in ovaries of the indicated genotypes. **e** Bar plot showing the percentage of stage 10 follicles with *ZAM* RNA detected in the ooplasm by smRNA FISH of RevI-H2i2 ovaries with the indicated GLKD. This experiment was done four times resulting in a total number of $n = 172$ for RevI-H2i2 *white*-GLKD, $n = 105$ for RevI-H2i2 *vret*-GLKD, $n = 160$ RevI-H2i2 *ago3*-GLKD follicles, $n = 168$ for RevI-H2i2 *armi*-GLKD, $n = 172$ for RevI-H2i2 white GLKD, $n = 109$ for RevI-H2i2 *zuc*-GLKD, $n = 97$ for RevI-H2i2 *piwi*-GLKD independent stage 10 follicles. In the box plots, the midline corresponds to the median value; the lower and upper hinges correspond to the first and third quartiles; and the whiskers span the minimum and maximum values, excluding outliers. *: $p$ value = 0.02857; ns, not significant ($p$ value = 0,8857) (Mann-Whitney, two-sided test). **f** Scatter plot showing the normalized counts of antisense Piwi-bound piRNAs mapping to individual internal TEs sequences in control ovaries (*white*-sKD) and *vret*-sKD ovaries. Antisense piRNA counts, mapped allowing up to 3 mismatches, were normalized per million of genome-mapping piRNAs (RPM, here in logarithmic scale). TEs in red have a *vret*-sKD/*white*-sKD ratio <0.3. *ZAM* is boxed black. **g** Confocal images of ovarioles showing *ZAM* smRNA FISH signal (in red) in *white*- (control) and v*ret*-sKD ovaries. DNA was stained with DAPI (white). **h** Color-inverted confocal projection of stage 10 egg chamber showing *ZAM* sm*RNA FISH signal in v*ret*-sKD ovaries. **i** Bar plot showing the percentage of stage 10 follicles with *ZAM* RNA invasion of the ooplasm, assessed by *ZAM* smRNA FISH, of *white*- (control) and *vret*-sKD ovaries. This experiment was done 4 times resulting in a total number of $n = 116$ *white*-sKD and $n = 98$ *vret*-sKD independent stage 10 follicles. In the box plots, the midline corresponds to the median value; the lower and upper hinges correspond to the first and third quartiles; and the whiskers span the minimum and maximum values, excluding outliers.

On the other hand, in the RevI-H2 line where *ZAM* was also reactivated, a large deletion of *flamenco* occurred during non-targeted mutagenesis and an adaptive response had already been set up in the germline. Thus, we chose to create a condition where a single TE is reactivated de novo and the piRNA pathway is fully functional.

Using CRISPR-Cas9, we de novo deleted the longest *ZAM* copy in the *flam* piRNA cluster in a line carrying the X chromosome of the Iso1A reference line. As in this line, the *flam ZAM* copy is at the genomic position X:21,778,810..21,783,994, we used two guides designed to create a deletion spanning over X:21,777,135..21,784,062 (6926 pb). We named the resulting line *flamΔZAM*. Mapping of genome-unique piRNAs to the *flam* locus highlighted the complete loss of piRNA production in the targeted region compared with the control Iso1A line (Fig. 4a). Conversely, the global production of genome-unique piRNAs mapping upstream and downstream of *ZAM* was not affected. We confirmed that the deletion induced a strong decrease of all piRNAs mapping to the internal regions of the reference *ZAM*. However, piRNAs targeting the *ZAM* LTR were still produced (Fig. 4b). In line with these results, PCR amplification and DNA sequencing of the CRISPR-Cas9 target locus showed that only the internal regions of *ZAM* had been deleted from the *flam* piRNA cluster, resulting in the retention of a solo-LTR at the initial *ZAM* insertion site (Supplementary Figure 4a).

In the *flamΔZAM line*, only the production of *ZAM*-derived antisense piRNAs was strongly altered, whereas the production of antisense piRNAs mapping to other TEs was not affected (Fig. 4c). This shows that the *ZAM* deletion from *flam* impairs *ZAM*-derived piRNA production, but does not affect the global piRNA production in ovaries. In sum, in the *flamΔZAM* line, the piRNA pathway was functional and almost no piRNAs that could target the internal *ZAM* sequences were produced.

We then analyzed *ZAM* RNA expression by smRNA FISH in *flamΔZAM* ovaries. Surprisingly, we did not detect any *ZAM* expression (Fig. 4d – panel 1). Based on this observation, we hypothesized that either there was no functional *ZAM* copy in the *flamΔZAM* line, or that the *ZAM* solo-LTR retained in *flam* was sufficient to silence *ZAM* expression. To check this, we exchanged chromosome II or III of the *flamΔZAM* line with chromosome II or III of the RevI-H2i2 line that each contains three or two recently transposed potentially functional *ZAM* copies, respectively. We made sure not to introduce the ZAM copies located in germline piRNA clusters of the X chromosome of the RevI-H2i2 line that are involved in the production of *ZAM*-regulating piRNAs in the germline. smRNA FISH revealed that *ZAM* RNAs were strongly expressed when these *ZAM* copies were added to the genome of the *flamΔZAM* line (Fig. 4d – panels 2 and 3). We also observed a strong invasion of oocytes with chromosome II, where approximately 70% of stage 10 follicles presented *ZAM* RNAs in the ooplasm, and moderate invasion with chromosome III. Similarly, we observed strong invasion of the oocyte when we added three euchromatic *ZAM* copies located on the telomere side of the X chromosome of the RevI-H2 line to the *flamΔZAM* genome (Fig. 4d – panel 4). Overall, our findings show that in the absence of *ZAM*-derived piRNAs in the somatic follicle cells and in the germline, transcripts from functional *ZAM* copies massively invade the oocytes. Additionally, our results demonstrate that the presence of a solo-LTR of *ZAM* (454 bp) in the *flam* piRNA cluster of the *flamΔZAM* line was not sufficient to silence *ZAM* expression in follicle cells (Supplementary Figure 4a). The three *ZAM* copies added to the X chromosome of the *flamΔZAM* line consist of one full length *ZAM* insertion (ZAM-fl), one ZAM-v2 and one ZAM-v3. Utilizing the sequence specificity of each insertion, we conducted RT-PCR analysis to determine which copies were expressed. The results revealed that ZAM-fl, ZAM-v2 and ZAM-v3 copies are expressed in RevI-H2i2 ovaries as well as when joined to the *flamΔZAM* genome (Supplementary Figure 4b).

An important observation was that the *flamΔZAM* flies that contain the three functional *ZAM* copies on the X chromosome displayed atrophied ovaries and reduced fertility compared with the control *flamΔZAM* flies that lack functional *ZAM* copies (Fig. 4e, f). Specifically, 90% of *flamΔZAM* females with active *ZAM* copies did not lay eggs. These data indicate that the reactivation of one single errantivirus in a patch of follicle cells can induce severe fertility defects. These results suggest that when *ZAM* piRNAs are produced in the germline, they contribute to ensure not only genome integrity but also fertility by protecting the oocytes against *ZAM* invasion.

## No initial small RNA response takes place after *ZAM* reactivation

The reactivation of an errantivirus, such as *ZAM*, in a genetic context where the piRNA pathway is functional provides a unique opportunity to examine the genomic immune response to TE reactivation and germline invasion. piRNAs and siRNAs are the two main classes of small RNAs produced to control the expression of transposable elements and counteract viral infection, respectively. Moreover, invading viral RNAs can be processed to small RNAs to initiate a primary response[40,41]. To characterize the small RNA response upon *ZAM* reactivation, we sequenced small RNAs that are complexed with Argonaute proteins (named here regulatory piRNAs) from *flamΔZAM* ovaries containing two or four potentially functional *ZAM* copies. In the *flamΔZAM* line, there were no de novo *ZAM* piRNAs (23 to 29 nt) produced upon *ZAM* reactivation, despite germline invasion. Indeed, only few sense and antisense *ZAM* piRNAs were detected in the ovaries of the *flamΔZAM* line that contained functional *ZAM* copies (Fig. 5a,

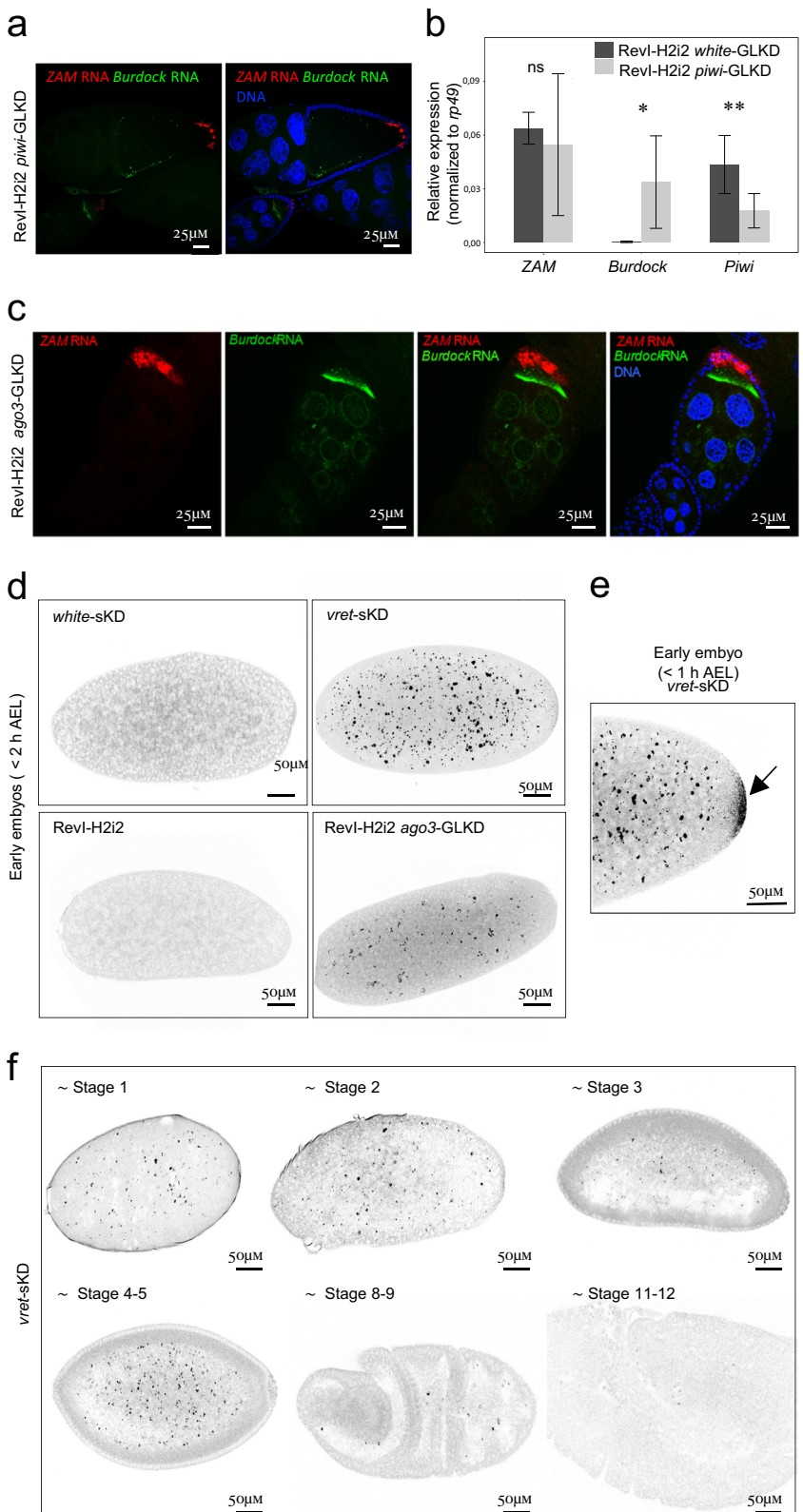

Supplementary Figure 5a). Moreover, this lack of piRNA production concerned only *ZAM*. The level of regulatory piRNAs targeting other TEs was not reduced (Fig. 5b, Supplementary Figure 5b).

We sequenced total small RNAs in *vret*- or *yb*-sKD ovaries where *ZAM* was also reactivated. Surprisingly we observed sense *ZAM*-mapping small RNAs of various sizes, including 23-29 nt small RNAs that are not produced in a *white*-sKD control condition (Supplementary

Figure 5c). However, the sense 23-29 nt *ZAM*-derived small RNAs detected in *yb*- and *vret*-sKD ovaries did not display uridine bias at the 5' end, a feature of mature primary piRNAs (Supplementary Figure 5d). Furthermore, these sense *ZAM*-mapping small RNAs were not immunoprecipitated with Piwi proteins. We detected almost no Piwi-bound sense or antisense *ZAM*-mapping piRNAs in *vret*- or *yb*-sKD ovaries (Supplementary Figure 5e). In addition, no sense regulatory *ZAM*

**Fig. 3 | *ZAM* RNAs transcribed in the follicle cells transit to the oocyte and are deposited in early embryos. a** Confocal images of stage 10 follicles in the RevI-H2i2 *piwi*-GLKD line. *Burdock* (green) and *ZAM* (red) mRNAs were detected by smRNA FISH. DNA was stained with DAPI (blue). The experiment was independently repeated at least 5 times, with similar results obtained each time. **b** Fold-change in the steady-state *ZAM*, *Burdock* and *Piwi* RNA levels for RevI-H2i2 *piwi*-GLKD ovaries compared with RevI-H2i2 *white*-GLKD ovaries (control), quantified by RT-qPCR (primer sequences in Supplementary Data 5). *n* = 3 biological independent samples and two technical replicates have been performed for each sample, resulting in a total of six data points per genotype. All data points are shown as dot plots. *: *p* value = 0,022, **: *p* value = 0,008; ns, not significant (*p* value > 0.05) (Mann-Whitney two-sided test). Data are presented as mean values and the error bars indicate the standard deviation from the mean values. **c** Confocal images of

ovarioles and of an isolate stage 8 egg chamber in the RevI-H2i2 *ago3*-GLKD line. *Burdock* (green) and *ZAM* (red) mRNAs were detected by smRNA FISH. DNA was stained with DAPI (blue). The experiment was independently repeated at least 5 times, with similar results obtained each time. **d** Color-inverted confocal images showing *ZAM* smRNA FISH signal in early embryos collected 0–2 hours after egg laying (AEL). Mothers were from the indicated genotype. The experiment was independently repeated at least 3 times, with similar results obtained each time. **e** Zoom on the posterior part of a *vret*-sKD early embryo collected at 0–1 hours AEL showing *ZAM* RNA accumulation at the posterior pole. The experiment was independently repeated at least 5 times, with similar results obtained each time. **f** Color-inverted confocal images showing *ZAM* smRNA FISH signal in embryos laid by *vret*-sKD females at the indicated developmental stages. The experiment was independently repeated at least 3 times, with similar results obtained each time.

piRNAs complexed with Argonaute proteins were detected in *vret*-sKD ovaries (Supplementary Figure 2f). These results suggest that the sense *ZAM*-mapping small RNAs are most likely degradation products of excess *ZAM* sense transcripts and not piRNAs. Thus, in these different conditions, we did not detect any primary piRNA response either in the somatic cells where *ZAM* is expressed or in the oocyte that it invades.

Given that the *ZAM* errantivirus shares some structural and functional characteristics with retroviruses, we also asked whether *ZAM* reactivation could trigger a siRNA response. For this, we analyzed the production of 21 nt small RNAs complexed with Argonaute proteins (regulatory siRNAs). Similar levels of *ZAM* 21 nt small RNAs were produced in a control line and in the *flamΔZAM* line with functional *ZAM* copies (Fig. 5c). Globally, the production of 21 nt small RNAs in this line was comparable to the control for all TEs, including *ZAM* (Fig. 5d). *ZAM*-mapping 21 nt small RNAs were slightly increased only in the *vret*-sKD condition, compared to the *white*-sKD control line without *ZAM* expression (Supplementary Figure 5f). Overall, our results suggest that no or very few siRNAs that could protect ovaries from *ZAM* activity are produced upon activation of this errantivirus in follicle cells.

Therefore, we propose that no innate small RNA response is triggered in the lines where *ZAM* is reactivated, suggesting that *ZAM* will not be brought under control until the establishment of specific adaptive immunity and immune memory.

## Discussion

The discovery that piRNAs produced in germ cells not only restrict TE expression in the germline itself but also counteract the invasion of errantiviruses from adjacent cells reveals a novel role of piRNAs produced locally as an effective defense mechanism at the tissue scale. When piRNAs against *ZAM* are produced in the germline, *ZAM* transcripts are limited to a patch of follicle cells and flies are fertile. Conversely, in the absence of *ZAM*-derived piRNAs in the germline, transcripts from functional *ZAM* copies massively invade the oocytes and are even transmitted to the embryos. At the molecular level, our study revealed that piRNAs produced in nurse cells and dumped into the oocyte can target RNAs produced by follicle cells and delivered to the oocyte. It is tempting to speculate that the *ZAM* RNAs arriving from somatic cells are targeted by complementary antisense piRNAs and degraded through the ping-pong cycle (reviewed in[33]). This would result in the production of *ZAM* sense piRNAs, thus participating in the amplification of the pool of piRNAs against *ZAM* and strengthening the defense mechanism against the invasion.

Transposition in the germ cell genome is crucial for TE propagation in a population because it allows the vertical transmission of new insertions. At the time of invasion (e.g. in the *flamΔZAM* line), no *ZAM*-derived piRNA was produced in somatic or germ cells. Therefore, this condition could be compared to what happens when a TE first enters a new species by horizontal transfer. In this case, according to several studies, an initial transposition burst occurs that leads to TE accumulation in the genome before the induction of an adaptive response by

the host to control transposition[42–47]. We found at least 17 new *ZAM* insertions in the RevI-H2i2 genome, testifying that *ZAM* actively transposed. Surprisingly, we identified three *ZAM* insertions in the same germline piRNA cluster on the X chromosome. It has been proposed that two different mechanisms can explain the de novo piRNA production required to specifically silence a novel invading TE. Indeed, piRNAs can be produced from a new TE insertion into a pre-existing piRNA cluster, and also by stand-alone TE insertions converted into piRNA-producing loci[48–51]. Interestingly, when active *ZAM* copies are added into a *flamΔZAM* genetic background, these copies do not produce regulatory *ZAM*-derived piRNAs and are not converted into piRNA producing regions. Actually, *ZAM* should be transcriptionally inactive in the germline because *ZAM* expression is regulated by the somatic transcription factor Pointed[34]. Therefore, although the contribution of individual copies needs to be investigated, our data strongly suggest that germline *ZAM* piRNAs originate at least in part from piRNA clusters. In line with our findings, computer simulations of the TE invasion dynamics suggested that several insertions in piRNA clusters are likely to be required to stop the invasion in the germline[52]. Conversely, we found that a single insertion in a somatic piRNA cluster, such as *flamenco*, is sufficient for TE silencing in the somatic follicle cells, as previously suggested[25]. Indeed, we show here that in follicle cells, the *flamenco* piRNA cluster acts alone as the principal regulator of transposon activity. A precise deletion of the internal part of the *ZAM* TE inserted in *flamenco* leads to complete derepression of functional *ZAM* copies. Surprisingly, it seems that reactivation of this errantivirus in only a patch of somatic cells can lead to severe fertility defects. However, in the germline, the deletion of the three most highly expressed germline piRNA clusters (42AB, 38 C, and 20 A) affected neither fertility nor TE silencing[50]. This could be explained by the fact that in the germline, but not in somatic cells, redundancy in piRNA production among different piRNA clusters is observed, with probably multiple piRNA clusters involved in the silencing of the same TE. Moreover, many TE copies have acquired mutations and are nonfunctional for transcription or for transposition. In the *flamΔZAM* line, although no *ZAM* piRNA was produced, we did not detect any *ZAM* expression. However, when we introduced functional copies of *ZAM*, *ZAM* was expressed demonstrating that depending on the genetic background, the transposon landscape varies and functional copies can be absent or present. In general, the TE content varies considerably among populations[49,53]. Interestingly, we identified *ZAM* copies with a deletion that affects the *pol* gene. The copy numbers of this *ZAM* variant ZAM-v2 and of *ZAM* full-length elements (ZAM-fl) that have transposed in the RevI-H2 line are similar (4 and 6 copies respectively), suggesting that even this *ZAM* variant can transpose very efficiently. Studies in different organisms have shown that TEs rapidly diversify. For instance, the *mariner* DNA transposon is present in 68 different versions in grass genomes[54]. Such rapid diversification certainly promotes speciation of new families of active TEs.

One interesting question is how *ZAM* RNAs transit into the oocyte and at which stage during *Drosophila* oogenesis *ZAM* transposes in

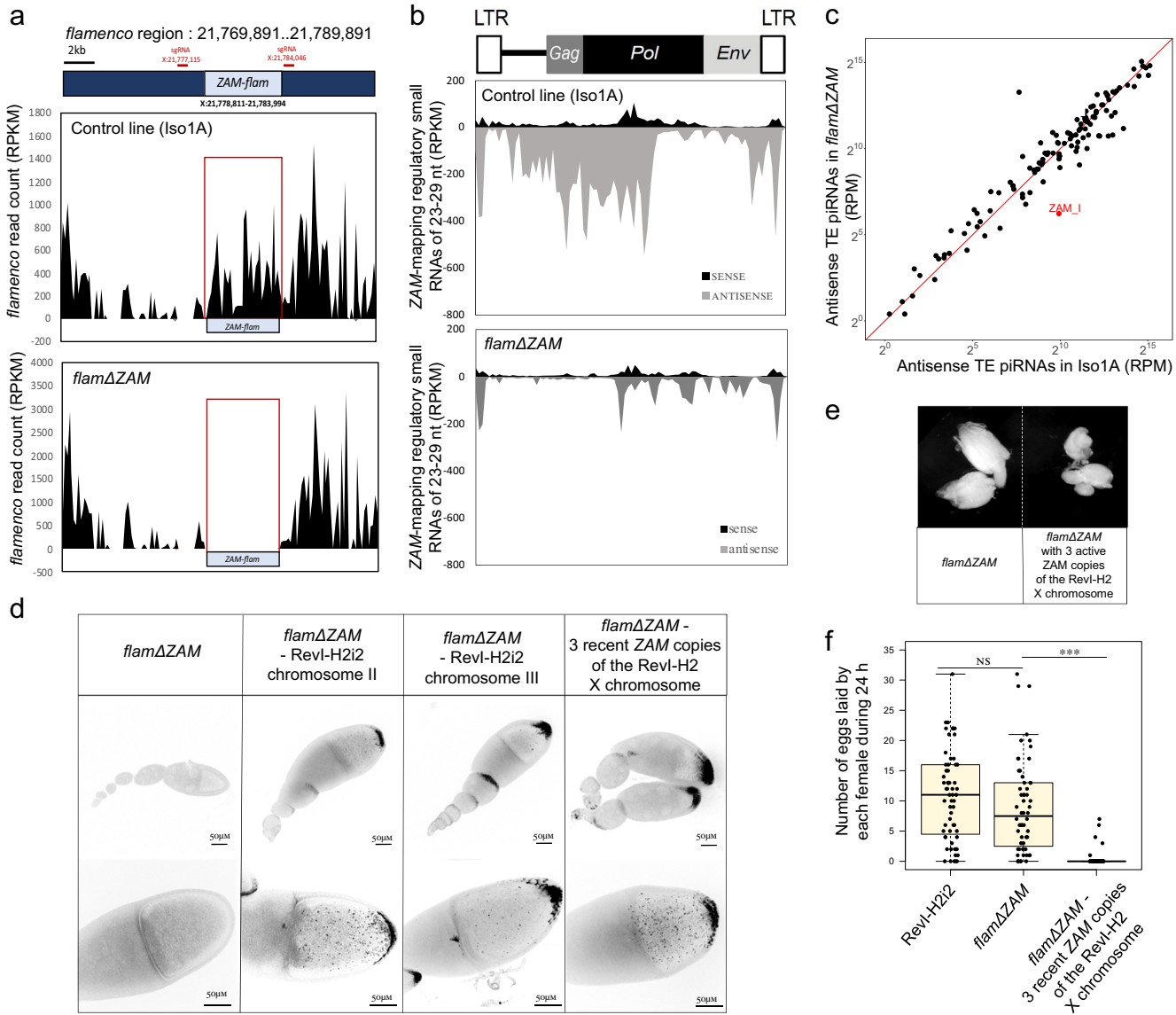

**Fig. 4 | *ZAM* deletion from the *flamenco* piRNA cluster leads to *ZAM* reactivation and oocyte invasion. a** Density plots showing genome-unique piRNAs mapping over 20 kb of the *flamenco* piRNA cluster (without mismatch) where the *ZAM* insertion is located (Release 6: X:21,769,891..21,789,891) in the control (Iso1A) and *flamΔZAM* lines. The position of the *flamenco ZAM copy (ZAM-flam)* is indicated. The positions of the sgRNAs used for *ZAM-flam* deletion by CRISPR-Cas9 are shown in red. **b** Density plot of *ZAM*-mapping regulatory piRNAs along the *ZAM* sequence in control and *flamΔZAM* ovaries (up to 3 mismatches). **c** Scatter plot showing the normalized counts of antisense regulatory piRNAs mapping to individual internal TE sequences in control ovaries (Iso1A) versus *flamΔZAM* ovaries. Antisense piRNA counts, mapped allowing up to 3 mismatches, were normalized per million of genome-mapping piRNAs (RPM, here in logarithmic scale). TEs in red have a *flamΔZAM*/Iso1A ratio <0.3. **d** Color-inverted confocal images of ovarioles (upper panels) and stage 10 egg chambers (lower panels) from the indicated genotypes showing *ZAM* smRNA FISH signal. The experiment was independently repeated at least 5 times, with similar results obtained each time. **e** Ovaries of the *flamΔZAM* line and of the *flamΔZAM* line with three recent *ZAM* copies introduced by genetic crossing and X chromosome recombination with the RevI-H2 line. **f** Box plot displaying the number of eggs laid per fly per day by RevI-H2i2, *flamΔZAM* and *flamΔZAM* females with three recent *ZAM* copies. Each dot represents an individual female. $n = 63$ for RevI-H2i2, $n = 60$ for *flamΔZAM*, $n = 60$ for *flamΔZAM* with three recent *ZAM* copies. In the box plots, the midline corresponds to the median value; the lower and upper hinges correspond to the first and third quartiles; and the whiskers span the minimum and maximum values, excluding outliers. ***$p$ value < 2.2e-16; ns, not significant ($p$ value > 0.05) (Mann-Whitney two-sided test).

germ cells. We noticed that *ZAM* RNAs, although transcribed in follicle cells, were deposited and accumulated at the posterior pole of early embryos, where future germ cells cellularize. *ZAM* might have evolved to preferentially mobilize in the dividing primordial germ cells of the offspring instead of the developing oocyte, which is not in a phase favorable to transposition because of its prolonged arrest in prophase I associated with highly compacted chromatin[55].

Previous experiments demonstrated that *ZAM* RNAs accumulate in follicle cells when vitellogenesis is defective, suggesting that *ZAM* transmission to the oocytes requires functional vitellogenin

trafficking[11] (Supplementary Movie 1). Moreover, errantiviruses, such as *ZAM* and *Gypsy*, can form pseudo-viral particles in follicle cells when they are expressed[8,10,11,15]. *ZAM* Gag and Env proteins are expressed in RevI-H2i2 follicle cells. Retroviral envelope glycoproteins undergo proteolytic processing by cellular serine endoproteases (furin and proprotein convertases), in order to produce the two functional subunits, a glycosylated hydrophilic polypeptide (SU) and a transmembrane domain (TM). This step is necessary to achieve protein competence to promote membrane fusion and virus infectivity[56,57]. The errantiviral *env* gene, acquired from insect baculovirus[2], encodes a

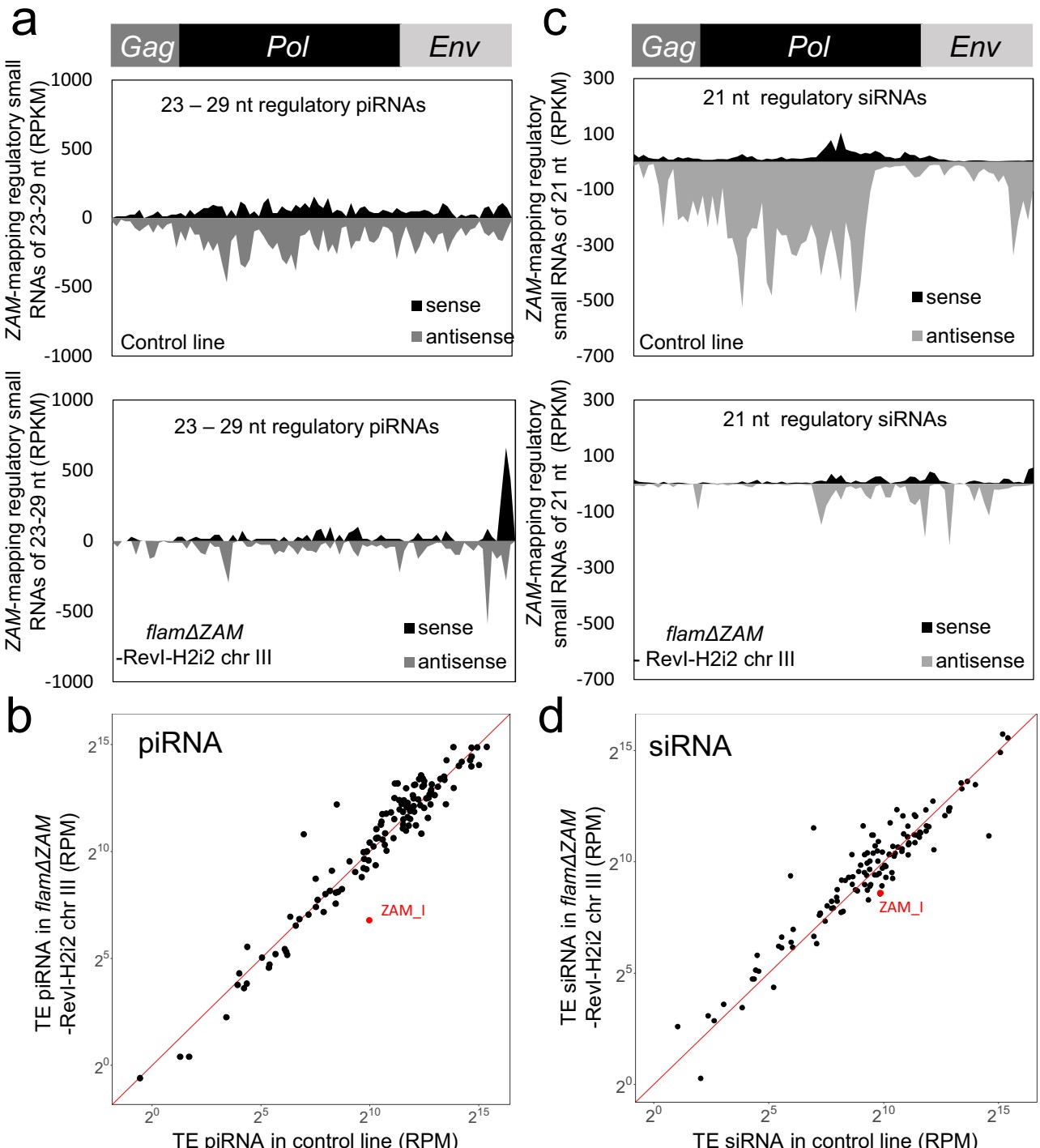

**Fig. 5 | No initial piRNA and siRNA response after *ZAM* retrotransposon reactivation. a** Density plot of regulatory piRNAs along the *ZAM* internal sequence in ovaries from a control line (*white*-sKD) and from the *flamΔZAM*-RevI-H2i2 chr III line (up to 3 mismatches). **b** Scatter plots showing the normalized counts of regulatory piRNAs mapping to individual internal TE sequences in control ovaries (*white*-sKD) versus *flamΔZAM*-RevI-H2i2 chr III ovaries. piRNA counts, mapped allowing up to 3 mismatches, were normalized per million of genome-mapping piRNAs (RPM, here in logarithmic scale). **c** Density plot of *ZAM*-mapping regulatory siRNAs (21nt small-RNAs complexed with Argonaute proteins) along the *ZAM* internal sequence produced in control (*white*-sKD) and in *flamΔZAM* RevI-H2i2 chr III ovaries (0-1 mismatch). **d** Scatter plot showing the normalized counts of antisense regulatory siRNAs mapping to individual internal TE sequences in control ovaries (*white*-sKD) versus *flamΔZAM* RevI-H2i2 chr III ovaries. Antisense siRNA counts, mapped allowing up to 1 mismatch, were normalized per million of genome-mapping siRNAs (RPM, here in logarithmic scale).

protein whose function is analogous to that of retroviral Env protein, which is responsible for infectious properties. Indeed, *ZAM* Env protein is composed of an SU, a TM and an Arg−X−Lys−Arg conserved domain, which is considered to be a furin consensus proteolytic cleavage site[28,57,58]. A recent study demonstrated that intracellular retrotransposition of the *Gypsy* errantivirus can occur in the absence of the Env protein, while intercellular transmission requires a functional Env[10]. Currently, whether *ZAM* can transpose intracellularly, in follicle cells, and the exact mechanisms (Env protein dependent or independent) used for *ZAM* transmission to the oocyte are unknown.

Once a virus enters the target cells, several scenarios for retrovirus uncoating have been proposed. Recent advances on the HIV-1 strongly suggest an uncoating at the nuclear pore and within the nuclear compartment[59]. Errantiviruses also encode Gag proteins which are capable of mediating the assembly of virus-like particles. However, there are significant differences between these proteins and retroviral Gag proteins, which might lead to a different uncoating process for errantiviruses[60]. In our study, we observed the loss of *ZAM* RNA in the oocyte when germline *ZAM*-derived piRNAs are produced. We can hypothesize that the *ZAM* capsid is dissociated soon after entry into the oocyte and that piRNAs can therefore easily target *ZAM* RNAs. It is also possible that *ZAM* RNAs are still encapsidated in the oocyte, but the capsid does not protect the *ZAM* RNA from piRNAs and their associated endonuclease. Previous studies on virus-like particles of the yeast *Ty* retrotransposon suggest that these particles form open structures that leave the RNA at the interior of the capsid accessible to RNase[61,62].

It has been proposed that in different species, an "innate" response at the piRNA or siRNA level can be initiated in the gonads upon retrovirus invasion ([41,63–68]). We found that no clear innate small RNA response is triggered upon *ZAM* reactivation in *Drosophila* ovaries. Total small-RNA sequencing after *ZAM* reactivation revealed the presence of sense *ZAM*-mapping small RNAs, but they lack characteristics of siRNAs or piRNAs. We hypothesize that these sense small RNAs are degradation products generated when there is an excessive amount of *ZAM* RNAs. Altogether, these results show that no initial small RNA response is mounted upon *ZAM* reactivation, leaving follicle cells, where *ZAM* is expressed, and also oocytes unprotected against *ZAM* transposition. Our findings suggest that the piRNA response is a robust response that, in the case of *ZAM*, must be established either in the follicle cells or in the germ cells to efficiently control the invading TE. However, the time required for developing a robust piRNA response to sustainably control TE transposition remains unknown. Lastly, it is thought that the piRNA pathway has no antiviral role in *Drosophila*[69]. However, it would be interesting to determine whether this is also true if parts of the viral genome are integrated into a germline piRNA cluster during infection.

## Methods

### Fly stocks, transgenic lines, and crosses

Flies were maintained at 25 °C under a light/dark cycle and 60% humidity. Between 3 and 6 days after hatching, flies were used for experiments. The RNAi lines against components of the piRNA pathway used in this study are listed in Supplementary Data 2. Isogenic lines from the RevI-H2 stock were generated by crossing a single female with a single male for five generations. Germline and somatic knock-down have been performed using the nanos-gal4 and the traffic-jam-gal4 driver lines respectively (Supplementary Data 2). The *Drosophila* line carrying the *ZAM* sensor is described in[30]. The sensor expresses the GFP reporter gene under the control of an inducible Upstream Activation Sequence promoter (UASp) and harbors a *ZAM* fragment in its 3′-UTR (pGFP-*ZAM*).

### Single-molecule RNA fluorescence in situ hybridization (smRNA FISH) in ovaries and embryos

All reagents are described in Supplementary Data 7. *ZAM* smRNA FISH was performed using 48 probes that target *ZAM* transcripts in a region that is absent in the *ZAM* inserted in the *flamenco* piRNA cluster to detect only transcripts of active *ZAM* copies (sequences in Supplementary Data 3). Ovaries from 3 to 6-day-old flies were dissected in Schneider's *Drosophila* Medium and fixed in Fixing Buffer (4% formaldehyde, 0.3% Triton X-100, 1x PBS) for 20 min at room temperature, rinsed three times in 0.3% Triton X-100, once in PBS, and permeabilized in 70% ethanol at 4 °C overnight. Permeabilized ovaries were rehydrated in smRNA FISH wash buffer (10% formamide in 2x

SSC) for 10 min. Ovaries were resuspended in 50 μL hybridization buffer (10% dextran sulfate, 10% formamide in 2x SSC) supplemented with 1 μL of smRNA FISH probes. Hybridization was performed with rotation at 37 °C overnight. Ovaries were then washed twice with smRNA FISH wash buffer at 37 °C for 30 min and twice with 2xSSC solution. Then, DNA was stained with DAPI (1/500 dilution in 2x SSC) at room temperature for 20 min. Ovaries were mounted in 30 μL Vectashield mounting medium and imaged on a Zeiss LSM-980 or Zeiss LSM-800 confocal microscope. The resulting images were processed using FIJI/ImageJ. The RNA signal specificity was confirmed by adding 1 mg/mL of RNase A in 2x SSC for 2 h before the hybridization step.

For embryo staining, flies were caged and fed yeast paste. Embryos (0–2 h) were collected, dechorionated in 50% bleach for 4 min and rinsed in water. Eggs were fixed in 4% paraformaldehyde/heptane for 20 min, devitellinized by vigorous shaking in 100% methanol and stored in methanol at −20 °C. Embryos were rehydrated with 1:1 methanol in 1xPBS, 0.1% Tween-20 for 5 min and twice in 1xPBS, 0.1% Tween-20. Embryos were resuspended in smRNA FISH wash buffer (10% formamide in 2x SSC) for 10 min and then processed for smRNA FISH as described for ovaries. Immunostaining combined with smRNA FISH was performed by adding primary antibodies to the smRNA FISH probes in the hybridization buffer and incubating at 37 °C overnight. Embryos were washed twice with smRNA FISH wash buffer and incubated with the secondary antibody for 90 min. After two washes in 2x SSC, embryos were mounted in 30 μL Vectashield mounting medium.

### Immunofluorescence analysis of ovaries and embryos

All reagents are described in Supplementary Data 7. Ovaries from 3–6-day-old flies were dissected in supplemented Schneider's medium, ovarioles were separated, and the muscle sheath was removed before fixation to obtain undistorted follicles. Then, ovaries were fixed in 4% formaldehyde/1x PBS/2% Tween-20 for 15 min, rinsed three times with 1x PBS/2% Tween-20, and incubated 2 hours in PBT (1x PBS, 0.1% Triton X-100, 1% BSA). Ovaries were incubated with primary antibodies in PBT at 4 °C overnight (antibodies are described in Supplementary Data 4). After three washes in PBT, ovaries were incubated with the corresponding secondary antibodies coupled to Alexa-488 or Cy3 for 90 min. After two washes in 1x PBS, DNA was stained with DAPI (1/500 dilution in 1x PBS) for 20 min. Ovaries were mounted in 30 μL Vectashield mounting medium and imaged on a Zeiss LSM-980 or Zeiss LSM-800 confocal microscope. The resulting images were processed using FIJI/ImageJ.

### RT-qPCR analysis of transposon expression

All reagents are described in Supplementary Data 7. 10-20 pairs of dissected ovaries were homogenized in TRIzol reagent. Following DNase I treatment, cDNA was prepared from 1 μg RNA by random priming of total RNA using Superscript IV Reverse Transcriptase. Quantitative PCR was performed with Roche FastStart SYBR Green Master on a the Lightcycler® 480 Instrument. RT-qPCR was used for quantification of transposon mRNA levels (primer sequences in Supplementary Data 5). All experiments were done with three biological replicates and with technical triplicates. Steady-state RNA levels were calculated from the threshold cycle for amplification with the $2^{-\Delta\Delta CT}$ method; *rp49* was used for normalization.

### Immunoprecipitation of Piwi-ribonucleoprotein complexes

All reagents are described in Supplementary Data 7. For each genotype, 50 pairs of ovaries from 3-6-day-old flies were dissected, lysed in 1 ml lysis buffer (20 mM HEPES-NaOH at pH 7.0, 150 mM NaCl, 2.5 mM MgCl₂, 250 mM sucrose, 0.05% NP40, 0.5% Triton X-100, 1x Roche-Complete). Samples were cleared by centrifugation at 10,000× *g* at 4 °C for 10 min. Extracts were incubated with rotation with rabbit polyclonal anti-Piwi antibodies (4 μg per sample) at 4 °C for 4 h

followed by overnight incubation with Dynabeads™ Protein A (50 µl, Invitrogen, 10002D) at 4 °C with rotation. Before incubation, beads were equilibrated in NT2 buffer (50 mM Tris-HCl, pH 7.4, 150 mM NaCl, 1 mM MgCl₂, 0.05% NP40). Beads were washed twice with ice-cold NT2 and twice with NT2 in which NaCl concentration was adjusted to 300 mM. Nucleic acids that co-immunoprecipitated with Piwi were isolated by treating beads with 0.7 mg/ml proteinase K in 0.3 ml proteinase K buffer (0.5% SDS, 10 mM Tris-HCl pH 7.4, 50 mM NaCl, 5 mM EDTA), followed by phenol/chloroform extraction (phenol at neutral pH) and ethanol precipitation.

### DNA Isolation, Oxford Nanopore Technology sequencing and genome analysis

All reagents are described in Supplementary Data 7. DNA was extracted from 200 RevI-H2i2 females using the Qiagen Blood & Cell Culture DNA Midi kit. The genomic DNA quality and quantity were evaluated using a Femto Pulse (Agilent) and a Qbit 3.0 (Invitrogen) respectively. Oxford Nanopore Technology (ONT) sequencing was performed by I2BC (Gif-sur-Yvette, France) using five micrograms of genomic DNA. Adapter-trimmed ONT reads were analyzed using NCBI Blastn (https://blast.ncbi.nlm.nih.gov/Blast.cgi) or command-line Blastn (Blastn 2.8.1). The RevI-H2i2 ONT reads containing ZAM insertions were detected by Blastn with the reference ZAM[22], Repbase ZAM_I and ZAM_LTR sequences (https://www.girinst.org/repbase/[31],). The ZAM containing reads were then recovered with bedtools getfasta (https://bedtools.readthedocs.io/en/latest/content/tools/getfasta.html[70],). The ZAM insertion sites where determined using Blastn of the ZAM containing reads to the D. melanogaster Release 6 genome (http://flybase.org). The absence of empty sites, without a given ZAM insertion, was verified by Blastn of the respective empty insertion sites, recovered from the D. melanogaster Release 6 genome, to all ONT reads. Sequencing data are available in NCBI GEO database, accession GSE213456.

### RNA-sequencing and analysis

3 independent total RNA extractions from 30 ovaries from 3-6-day-old RevI-H2i2 flies using Trizol (Invitrogen) were performed. Strand-specific libraries for RNA sequencing (RNA-Seq) were constructed at BGI. 1 µg of total RNA was used to prepare mRNA library using MGIEasy RNA Library Prep kit and sequenced as PE150 reads on the DNBSEQ G400 sequencer, and adapter-clipped reads were provided by BGI. The data were analyzed on the local Galaxy platform using HISAT2 alignment (Galaxy Version 2.1.0). The reads were first mapped to ZAM-fl and to ZAM-v2 using the "Disable spliced alignment" option. The mapped reads were then aligned to ZAM_I sequence (Repbase) with the default options, i.e. allowing split alignment. Reads were visualized using "Integrative Genomics Viewer" version 2.16.1. The coverage of RNA-seq reads corresponding to ZAM-v2 was determined as the number of split reads specific to the ZAM-v2 deletion at positions ZAM_I:5494-6120, normalized to the average coverage of the pointed transcript. Coverage of RNA-seq reads corresponding to ZAM-fl and ZAM-v2 was determined as the number of nonsplit reads at position 5495, the left edge of the deletion, normalized to the average coverage of the pointed transcript. Only the left edge was used for this purpose, as the right edge of the deletion is also mapped by reads corresponding to ZAM variants with other internal deletions (green in Fig. 1a). RNA sequencing data are available in NCBI GEO database, accession GSE235966.

### Small RNA sequencing

Regulatory small RNA extraction was performed as described in[71]. Briefly, 50 pairs of ovaries from 3-6-day-old flies were lysed and Argonaute-small RNA complexes were isolated using TraPR ion exchange spin columns (Lexogen, Catalog Nr.128.08). Total RNA also was extracted for some samples (80-100 ovaries for each sample) using the classical TRIzol extraction, followed by 2 S rRNA depletion and size selection before sequencing. Sequencing was performed by Fasteris SA (Geneva, Switzerland) on an Illumina NextSeq550 instrument (13–15 million reads per sample).

Illumina small RNA sequencing reads were loaded on the small RNA analysis pipeline sRNAPipe[72] for mapping to the various genomic sequence categories of the D. melanogaster genome (Release 6). For the analysis, 23-29nt genome mappers were selected as piRNAs, and 21nt genome mappers were selected as siRNAs. Genome-unique piRNAs were defined as 23–29 reads that mapped uniquely across the reference genome. piRNAs were mapped to TEs allowing up to 3 mismatches, siRNAs allowing up to 1 mismatch, and genome-unique piRNAs to piRNA clusters as defined in[22] allowing no mismatch. The window size was of 91nt for the entire ZAM, 86 nt for ZAM_I and 140nt for the flamenco region at X:21,769,891..21,789,891 to establish the piRNA density profiles. For comparison between samples, all read counts were normalized by the number of piRNA/siRNA reads used as input (genome-mapped piRNAs or siRNAs, genome-unique piRNAs) and represented in RPM [RPM = read-count * 1,000,000 / (total of genome mapping piRNAs)], or RPKM [RPM = read-count * 1,000,000 / (TE Length/1000) / (total of genome mapping piRNAs)] in the case of density profiles. To assess the ping-pong signature, the counts of 1 to 23nt 5'-overlaps were determined, and the percentage of 10nt 5'-overlaps over the total 1-23nt 5'-overlaps was calculated. The Z-score for 10nt 5'overlaps was determined using the percentages of all 1 to 23nt-long 5'overlaps as a background and was considered significant when it was >1.96. Scatter plots were done with RStudio with antisense piRNAs or siRNAs mapping to the respective internal sequence of TEs (allowing 0-3 mismatches for piRNAs and 0-1 mismatch for siRNAs). All small RNA sequencing done for this study is listed in Supplementary Data 6. Small-RNA sequencing data are available in NCBI GEO database, accession GSE213368.

### Sterility tests

For the sterility tests, RevI-H2i2 homozygous females were compared to w[1118] females, and homozygous flamΔZAM females that carry three functional ZAM copies from the RevI-H2 X chromosome were compared to flamΔZAM females without these functional ZAM copies in their genome. Sterility tests were performed at 25 °C. Thirty 3-6-day-old virgin females of each genotype were individually mated with three w[1118] males, and eggs were collected for 24 hours. The number of eggs laid by each female was determined. Then, eggs were kept at 25 °C for another 24 hours before determining the egg hatching rate. The experiment was done twice for a total of approximately 60 replicates for each condition.

### Generation of the flamΔZAM line using the CRISPR-Cas9 technology

The ZAM copy in the flamenco piRNA cluster was deleted using a pCFD6 plasmid that expresses two single guide RNAs (sgRNAs) under the Gal4/UAS system. The sgRNAs were designed to target genome-unique sequences located upstream and downstream of the flamenco ZAM copy: TTGTAGCGCTCTTCTTCTCT (sgRNA flamΔZAM_1) and AGCGCAACCACGTACAGCGA (sgRNA flamΔZAM_2). The pCFD6 plasmid harboring these sgRNAs was injected by the Bestgene company into embryos from the BL#9736 stock (Bloomington stock center) for integration of the plasmid into the genomic site 53B2. The obtained flies were crossed with nanos-gal4; UAS-Cas9 flies (BL#54593, Bloomington stock center). Prior to crossing, the X chromosome of each of these lines was replaced by the X chromosome of the Iso1A Drosophila line. After the crosses, 500 lines were derived and screened by PCR using primers specific for the flamenco ZAM copy. The detailed crossing schemes are available upon request. Lines without amplification of internal flamenco ZAM sequences were selected for PCR

amplification of the target locus with primers that framed the *flamenco ZAM* copy and the target sequences of the sgRNAs. The obtained amplicons were sequenced.

## Reporting summary

Further information on research design is available in the Nature Portfolio Reporting Summary linked to this article.

## Data availability

All sequencing data generated in this study have been deposited in NCBI GEO database, Sequence Read Archive, under accession code GSE213456 for the ONT genome sequencing data, GSE213368 for the small-RNA sequencing data and GSE235966 for the RNA sequencing data. Source data are provided as a Source Data file. Source data are provided with this paper.

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

## Acknowledgements

We thank the *Bloomington Stock Center* for providing stocks. We thank J. Brennecke for providing antibodies. We thank www.flybase.org, http://www.girinst.org and NCBI for providing databases and tools. We thank S. Chambeyron and K. Senti for helpful discussions, A. Molaro and the members of the Brasset laboratory for comments on the manuscript. We thank C. Vaury for having established the original RevI-H2 line over 30 years ago, and for preserving it all these years. We thank Y. Renaud and P. Pouchin for helpful advices and development of the iGReD bioinfomatics platform. We also thank the CLIC facility (Clermont Imagerie Confocale). We thank Gentyane Facility for the Quality Control of gDNA. This work was supported by grants from the Agence Nationale pour la Recherche (CHApiTRE ANR-20-CE12-0005, EB, BiopiC ANR-21-CE12-0022, EB, and EpiTET ANR-17-CE12-0030-03, EB), the La Fondation ARC pour la recherche sur le cancer (PJA20171206129, EB). M.Y. was supported by the Ministère de l'Enseignement Supérieur et de la Recherche and the Fondation pour la Recherche Médicale (FDT202106012950, MY). This research was financed by the French government IDEX-ISITE initiative 16-IDEX-0001 (CAP 20-25), EB.

## Author contributions

Conceptualization, M.Y., S.J. and E.B.; Validation and Investigation, M.Y., S.M.M., A.A., N.G., B.B., Data curation and bioinformatic analyses, M.Y., S.J., writing—original draft, M.Y. and E.B.; writing—review and editing, M.Y., S.J. and E.B.; Supervision, S.J. and E.B., funding acquisition, M.Y. and E.B.

## Competing interests

The authors declare that no competing interest exists.
