## [Peer Review File · Nature Communications]

Reactivation of an errantivirus in *Drosophila* ovarian somatic tissue: from germline invasion to tamingReviewers' Comments:

Reviewer #1:

Remarks to the Author:

Yoth et al have performed a series of detailed experiments investigating host repression of the *Drosophila* ZAM LTR retrotransposon in somatic gonadal cells and the germline. This is an interesting topic and the authors appear to provide several interesting novel insights, aided by use of an impressive set of molecular approaches.

First and foremost, however, an important consideration is that the sequence Yoth et al focus on is incorrectly referred to as an 'endogenous retrovirus' and there is a mis-placed focus on retroviral terminology throughout the manuscript. While LTR retrotransposons and retroviruses are both reverse transcribing selfish genetic elements, the distinction between them is real and important (see the section 'LTR retrotransposons are not retroviruses' below). Consequently, in my view, this manuscript requires extensive and careful revision to shift the current focus away from retroviruses, and re-couch the study in terms of LTR retroelements.

I felt that the manuscript could also potentially be strengthened by a more organised approach towards the specific questions under investigation. Perhaps the specific questions being addressed could be posed explicitly in the introduction (instead of the more general list of outstanding questions), and dealt with in turn in the results section. In my view it would help to add clearer signposting of what is already known, and what the new contributions arising from this study are and their connotations, towards furthering understanding of the repression dynamics of LTR retrotransposons in somatic cells and the germline. Meanwhile, the distinctions between germline vs gonadal somatic cells are not explicitly described, and further description could be useful for non-specialist readers. I hope the authors find these suggestions useful towards strengthening their work.

LTR retrotransposons are not retroviruses:

Retroviruses are members of the family Retroviridae, a monophyletic group of viruses that infect vertebrates (<https://ictv.global/report/chapter/retroviridae/retroviridae>). Meanwhile, ZAM is a member of the genus *Errantivirus*, which belongs to the family *Metaviridae*, members of which are commonly referred to as Ty3/Gypsy LTR retrotransposons (<https://ictv.global/report/chapter/metaviridae/metaviridae/errantivirus>). ZAM possesses an envelope-like gene, but it is a Ty3/Gypsy LTR retrotransposon, not a retrovirus. Enveloped LTR retrotransposons such as ZAM could potentially be considered a form of endogenous virus, but not endogenous retroviruses. Thus, using the term 'retrovirus' to describe ZAM is wrong and generates confusion, and could be likened to referring to a lizard as a 'mammal'. Consequently, I strongly suggest that the authors do not refer to the ZAM Ty3/Gypsy LTR retrotransposon as a retrovirus, here or elsewhere, and that they retract previous incorrect usages to avoid causing unnecessary confusion. I realise that Leblanc et al (<https://doi.org/10.1093/emboj/16.24.7521>) referred to ZAM as an 'invertebrate retrovirus' in the title of the 1997 paper when describing ZAM. But this has propagated confusion. Elsewhere in the same manuscript Leblanc et al use the term 'retrovirus-like'. However, this usage should also be discouraged for the same reasons. Leblanc et al's use of the term 'gypsy-like' is more accurate, although note that ZAM appears to form a separate monophyletic clade to Gypsy LTR retrotransposons (also see Wei et al <https://osf.io/fma57/> regarding to the problematic nature of this name more generally).

Some other specific comments:

Lines 441-445: This section appears out of date. I suggest citing a more up to date review (if this is kept), as my understanding is that new insights have been uncovered regarding nuclear entry of the HIV-1 capsid via the nuclear pores over recent years. Additionally, this sentence should be re-phrased: "Capsid disassembly of endogenous retroviruses has not been investigated yet". ERVs are retroviral sequences that have become fixed in the host genome (via integration into the germline and

inheritance by subsequent host generations), and as such, they do not have capsids. Where ERVs retain the capacity to form new infectious virions, their mechanisms of entry into new host cells are the same as those of exogenous retroviruses.

Line 433: "The ZAM env gene is similar to the retroviral env gene, which is responsible for the infectious properties". Note that while retrovirus and LTR retrotransposon envelope genes are largely analogous in function, their sequences differ considerably, and they are not considered homologous (i.e. one of many biological distinctions between retroviruses and LTR retrotransposons).

Line 126: "deleted 5'-UTR" do the authors mean a deleted 5' LTR?

Line 19: "Endogenous retroviruses (ERVs) are transposable elements (TEs) related to infectious retroviruses". This description suggests that ERVs are a separate group of TEs that are somehow related to retroviruses. Instead, ERVs are direct copies of individual retrovirus genomes that have integrated in the host germline.

Line 45: "ERVs can only be maintained in a species if they are able to transpose into the germline genome". An ERV only becomes an ERV when it integrates in the germline, and it is maintained in a species until it degrades through mutational processes or is edited out through recombination (or when the host lineage carrying it is lost). I think what the authors are getting at is that the LTR retrotransposon lineage in question can only be maintained over the longer term across many host generations if it retains intact germline copies, which is typically achieved via additional instances of germline integration.

Line 52: "While their presence in genomes can be evolutionarily beneficial". Bear in mind that it is generally considered that the vast majority of TE integrations are negative or at best nearly-neutral with regards to the host. Only a tiny proportion of integrations are likely to be beneficial to the host in any way.

Line 28: when referring to the process of TE repression, the authors state that "Our results not only highlight how ERVs and their host adapt to each other"- how is the TE adapting to the host under this scenario?

Reviewer #2:

Remarks to the Author:

In this manuscript, Marianne et al., leveraged the RevI-H2 Drosophila line to study the reactivation of ZAM. The RevI-H2 derived isogenic line, RevI-H2i2, has a deletion (including ZAM sequence) in the major somatic piRNA cluster, resulting in ZAM derepression in follicle cells. Meanwhile, the RevI-H2i2 carries ZAM insertions into the germline piRNA clusters. Thus, RevI-H2i2 expresses germline piRNA to thwart ZAM invasion from follicle cells. By using CRISPR-Cas9, the ZAM specific deletion in flam results in ZAM invasion to oocytes and reduced female fertility. Since it is known that piRNAs are the major force to silence transposon in both somatic and germline cells of gonads, the selling point of this manuscript is the adaptive genomic immune response against the invading ERV. To strengthen it, the author should provide more evidence: 1. The follicle cells produced ZAM are capable of mobilizing into the genome of oocyte; 2. ZAM could insert into piRNA cluster, ultimately producing piRNA.

Major:

In Figure 1, although the ZAM transcripts accumulated about 3-fold in the follicle cells of RevI-H2i2 females compared to w1118(Figure 1B, C), this stain is stable (Line 142-145, Figure 1E, F). Given RevI-H2i2 harbors both full-length and truncated copies of ZAM (Figure 1A), it is better to distinguish whether the accumulated ZAM transcripts are full-length or truncated by Nanopore RNA sequencing.

Because only the full-length copies of ZAM are potentially mobile.

The RevI-H2i2 expresses germline piRNAs targeting ZAM (Figure 2A). Why there is about 10% of stage 10 cyst still have ZAM RNA in the ooplasm of white-GLKD (Figure 2E)?

It is a great idea to generate a single deletion of ZAM in flam (flam Δ ZAM, Figure 4A-C). Strikingly, the flam Δ ZAM carrying functional ZAM copies displayed atrophied ovaries, invaded ZAM transcripts, and 90% females are sterile (Figure 4E-F). Is this due to the transposition of ZAM in oocyte? If so, please quantify the new insertions.

Although the RevI-H2i2 line happens to have the germline expressed piRNAs to prevent ZAM invasion, there is no detectable de novo piRNA being produced in flam Δ ZAM (Figure 5A, B). How long does it take for the flies developing "adaptive genomic immune response against the constantly invading ERV"? It is better to keep raising the fertile flam Δ ZAM-RevIH2i2 flies, evaluating their fertility and sequencing the ZAM piRNAs generation by generation.

Minor point

The *Drosophila* and gene name (flam, w1118...) should be in italics.

Reviewer #3:

Remarks to the Author:

A class of gypsy retrotransposons has acquired an envelope gene to behave like a retrovirus in *Drosophila* species. It is known that these so-called errantivirus or envelope-carrying gypsy elements, including ZAM, are expressed in the ovarian somatic cells and infect oocytes. Thus, a distinct piRNA-based defence mechanism is in place in the ovarian somatic cells to suppress their expression. Authors previously showed that, a *Drosophila* strain that lacks the region that produces piRNAs against ZAM in the soma has acquired new insertions of ZAM elsewhere in the genome. They showed that those newly acquired insertions of ZAM produce piRNAs in the germline. In the current manuscript, Yoth et al demonstrated that those germline piRNAs target ZAM mRNAs that are expressed in the soma and transmitted to the germline.

Several studies have examined piRNA-based adaptation processes in *Drosophila* after an acquisition of new transposons. But the current study is unique in a sense that the niche where transposons are activated is different from the cells where the silencing occurs. Therefore, it adds a significant value to the existing body of literature. I recommend a publication in *Nature Communications* should the following comments be adequately addressed.

1. There is evidence that somatic gypsy elements turn into germline transposons over time. Can you rule out that some of new ZAM insertions in the RevI-H2 are now expressed in the germline, and those are the ones that are targeted by the germline-expressed piRNAs?

Along this line, there are weak but significant nuclear FISH signals of ZAM mRNA in the nurse cells even in the control knockdown in Figure 2. Are they reproducible? Are they not seen in the control strain?

2. Authors showed that a germline knockdown of Piwi does not derepress ZAM in the RevI-H2 background (Figure 2). I find this data point crucial because it supports the idea that the germline piRNAs target ZAM mRNAs from the soma.

However, I do not see the validation of this RNAi line. Can you show that other germline transposons are derepressed in the same cross?

3. Please outline individual egg chambers in FISH images when the borders are not clear. Images contain information of where in each egg chamber ZAM is expressed. It is hard to work it out without seeing the structure of individual egg chambers. This applies, for example, to Figure 1B.

4. The antibody staining of ZAM proteins in Figure 1D is not very clear. Please include a staining of control egg chamber that does not express ZAM to demonstrate specificities.

5. Likewise for Figure S2A, please include a control ovariole where the pGFP-ZAM reporter is suppressed in the soma.

6. Please make available the code used for the computational analyses. For example, it is not indicated how the RPKM was calculated for the data presented in Figure 2B. For this reason, I cannot evaluate the differences between different TEs. In fact, it appears that the piRNAs against the somatic transposon Gypsy10 have decreased upon knockdown of ago3 or zucchini in the germline.

7. It is helpful to indicate which ZAM insertions are thought to be active and which are producing piRNAs in the germline. I appreciate that one cannot nail down to individual insertions. However, the authors' previous publication showed that the X chromosome is responsible for making the germline piRNAs. This information should be clearly noted in this current manuscript. Otherwise, the experiments using flam-delta-ZAM + RevI-H2 chromosomes cannot be interpreted. For the same reason, please describe all the crosses that partially include RevI-H2i2 chromosomes to the chromosome level. For example, the RNAi lines used in the study are on different chromosomes. The testing flies may contain different combinations of RevI-H2i2-derived chromosomes depending on the crossing scheme.

We would like to thank the three reviewers for their critical and helpful comments and suggestions. We are confident that we addressed all points raised by the reviewers in the revised submission. We provide a point-by-point response to the comments and highlighted in color the changes made to the manuscript. There is no doubt that these changes have clearly improved the manuscript. We are very pleased that the three reviewers share our enthusiasm for the results described in our manuscript and note that our work provides interesting new perspectives and adds significant value to the existing literature. Please note that due to the work performed during the revision process Abdou Akkouche has been added to the list of co-authors and Stephanie Maupetit Mehouas who was the fourth author initially is now the second author, all authors agreed to the changes to the list of co-authors.

REVIEWER COMMENTS

Reviewer #1 (Remarks to the Author):

Yoth et al have performed a series of detailed experiments investigating host repression of the *Drosophila* ZAM LTR retrotransposon in somatic gonadal cells and the germline. This is an interesting topic and the authors appear to provide several interesting novel insights, aided by use of an impressive set of molecular approaches.

First and foremost, however, an important consideration is that the sequence Yoth et al focus on is incorrectly referred to as an ‘endogenous retrovirus’ and there is a mis-placed focus on retroviral terminology throughout the manuscript. While LTR retrotransposons and retroviruses are both reverse transcribing selfish genetic elements, the distinction between them is real and important (see the section ‘LTR retrotransposons are not retroviruses’ below). Consequently, in my view, this manuscript requires extensive and careful revision to shift the current focus away from retroviruses, and re-couch the study in terms of LTR retroelements.

We thank the reviewer for this valuable feedback. As detailed below, we have now taken great care, throughout the manuscript, to differentiate between retroviruses and errantviruses, a group of insect LTR retrotransposons. We paid attention to the nuances and intricacies of each type. We have relocated the focus in the introduction to ensure that it provides accurate and informative overview of the subject matter. We believe that these revisions have significantly improved the manuscript and have made it more precise.

I felt that the manuscript could also potentially be strengthened by a more organised approach towards the specific questions under investigation. Perhaps the specific questions being addressed could be posed explicitly in the introduction (instead of the more general list of outstanding questions), and dealt with in turn in the results section. In my view it would help to add clearer signposting of what is already known, and what the new contributions arising from this study are and their connotations, towards furthering understanding of the repression dynamics of LTR retrotransposons in somatic cells and the germline.

As suggested by the reviewer, we have removed the questions listed at the end of the introduction and now ask the questions along the introduction as gaps in knowledge appear.

Meanwhile, the distinctions between germline vs gonadal somatic cells are not explicitly described, and further description could be useful for non-specialist readers. I hope the authors find these suggestions useful towards strengthening their work.

We thank the reviewer for pointing out this omission. We have added a sentence in the introduction describing the structure of the *Drosophila* ovary.

LTR retrotransposons are not retroviruses:

Retroviruses are members of the family Retroviridae, a monophyletic group of viruses that infect vertebrates (<https://ictv.global/report/chapter/retroviridae/retroviridae>). Meanwhile, ZAM is a member of the genus Errantivirus, which belongs to the family Metaviridae, members of which are commonly referred to as Ty3/Gypsy LTR retrotransposons (<https://ictv.global/report/chapter/metaviridae/metaviridae/errantivirus>). ZAM possesses an envelope-like gene, but it is a Ty3/Gypsy LTR retrotransposon, not a retrovirus. Enveloped LTR retrotransposons such as ZAM could potentially be considered a form of endogenous virus, but not endogenous retroviruses.

We agree with the reviewer that we referred to the LTR retrotransposon ZAM as a *Drosophila* endogenous retrovirus (ERV) by analogy with retroviruses that have successfully colonized mammalian germline genomes. Indeed, the classification of transposable elements is complicated. Since the first classification proposed by D.J. Finnegan in 1989, an increasing number of complex and large eukaryotic genomes have been sequenced and annotated, and several revised classifications have been proposed (Boeke and Stoye 1997; Capy 2005; Wicker et al. 2007; Llorens et al. 2020).

As the reviewer mentioned, ZAM belongs to the genus of errantiviruses, a group of insect endogenous LTR elements that share structural and functional characteristics with endogenous retroviruses of vertebrates. As such, like *Gypsy* or *Tirant*, ZAM is compared in numerous publications to an ERV. To be precise in terminology, we followed the advice of the reviewer and explained in the introduction that ZAM is a member of the genus Errantivirus, which belongs to the family Metaviridae.

Thus, using the term ‘retrovirus’ to describe ZAM is wrong and generates confusion, and could be likened to referring to a lizard as a ‘mammal’. Consequently, I strongly suggest that the authors do not refer to the ZAM Ty3/Gypsy LTR retrotransposon as a retrovirus, here or elsewhere, and that they retract previous incorrect usages to avoid causing unnecessary confusion. I realise that Leblanc et al (<https://doi.org/10.1093/emboj/16.24.7521>) referred to ZAM as an ‘invertebrate retrovirus’ in the title of the 1997 paper when describing ZAM. But this has propagated confusion. Elsewhere in the same manuscript Leblanc et al use the term ‘retrovirus-like’. However, this usage should also be discouraged for the same reasons. Leblanc et al’s use of the term ‘gypsy-like’ is more accurate, although note that ZAM appears to form a separate monophyletic clade to Gypsy LTR retrotransposons (also see Wei et al <https://osf.io/fma57/> regarding to the problematic nature of this name more generally).

In answer, when referring to ZAM or other errantiviruses, we replaced "ERV" with "errantivirus" or "TE" and changed the title and passages of the manuscript in consequence, according to the reviewer's comments.

Some other specific comments:

Lines 441-445: This section appears out of date. I suggest citing a more up to date review (if this is kept), as my understanding is that new insights have been uncovered regarding nuclear entry of the HIV-1 capsid via the nuclear pores over recent years. Additionally, this sentence should be re-phrased: “Capsid disassembly of endogenous retroviruses has not been investigated yet”. ERVs are retroviral sequences that have become fixed in the host genome (via integration into the germline and inheritance by subsequent host generations), and as such, they do not have capsids. Where ERVs retain the capacity to form new infectious virions, their mechanisms of entry into new host cells are the same as those of exogenous retroviruses.

As suggested by the reviewer we have cited a recent review and eliminated the 3 hypothesis that was previously proposed in the literature. The paragraph has been re-phrased (lines 472-477).

“Once a virus enters into the target cells, several scenarios for retrovirus uncoating have been proposed. Recent advances on the HIV-1 strongly suggest an uncoating at the nuclear pore and within the nuclear compartment (Guedán et al. 2021). Errantiviruses also encode Gag proteins that are capable of mediating the assembly of virus-like particles. However, there are significant differences between these proteins and retroviral Gag proteins, which might lead to a different uncoating process for errantiviruses (Syomin et al. 1993).”

Line 433: “The ZAM env gene is similar to the retroviral env gene, which is responsible for the infectious properties”. Note that while retrovirus and LTR retrotransposon envelope genes are largely analogous in function, their sequences differ considerably, and they are not considered homologous (i.e. one of many biological distinctions between retroviruses and LTR retrotransposons).

As the reviewer correctly pointed out, the term "similar" is not appropriate to describe the relationship between retroviral and errantiviral envelope proteins. Our aim was to emphasize the conservation of the envelope protein's function, but we acknowledge that the sequence divergence between retroviral and errantiviral env genes is significant, as noted by the reviewer. It has been suggested that the errantiviral env gene was acquired from the genome of baculoviruses through recombination events, which explains why it is not homologous to retroviral env genes (Stefanov, Salenko, and Glukhov 2012).

We changed the sentence (lines 462-464) to “The errantiviral env gene, acquired from insect baculovirus (Malik, Henikoff, and Eickbush 2000), encodes a protein whose function is analogous to that of retroviral Env protein, which is responsible for infectious properties.”

We also changed, in line 83-84, to "ZAM has 3 intact ORFs encoding proteins whose function is analogous to that of the Gag, Pol and Env proteins of ERVs or retroviruses."

Line 126: “deleted 5’-UTR” do the authors mean a deleted 5’ LTR?

We do refer to 5’-UTR. ZAM has a 5’-UTR which can harbor deletions depending of the ZAM variants that are considered. Indeed, UTR had not been defined in the manuscript and thus we changed to "... one, named *ZAM-vI*, harbored a deleted 5'-UTR (5' untranslated region) ..." (lines 130-131).

Line 19: “Endogenous retroviruses (ERVs) are transposable elements (TEs) related to infectious retroviruses”. This description suggests that ERVs are a separate group of TEs that are somehow related to retroviruses. Instead, ERVs are direct copies of individual retrovirus genomes that have integrated in the host germline.

We changed this part of the Summary accordingly to this and other comments of reviewer 1: "Most *Drosophila* transposable elements (TEs) are LTR retrotransposons, some of which belong to the genus *Errantivirus* and share structural and functional characteristics with vertebrate endogenous retroviruses (ERVs)."

Line 45: "ERVs can only be maintained in a species if they are able to transpose into the germline genome". An ERV only becomes an ERV when it integrates in the germline, and it is maintained in a species until it degrades through mutational processes or is edited out through recombination (or when the host lineage carrying it is lost). I think what the authors are getting at is that the LTR retrotransposon lineage in question can only be maintained over the longer term across many host generations if it retains intact germline copies, which is typically achieved via additional instances of germline integration.

In fact, this paragraph not only concerns ERVs but all TEs. We thus changed this paragraph to refer to TEs in general, rather than just ERVs. We wanted to point out that TEs don't behave as classical genes. The function of TE genes is to ensure transposition. Over time, TEs accumulate mutations and they risk becoming non-functional copies. But as long as a functional copy remains and transposes from time to time in the germline to create new functional copies transmitted to the next generation, the deleterious effect of mutations can be antagonized.

Line 52: "While their presence in genomes can be evolutionarily beneficial". Bear in mind that it is generally considered that the vast majority of TE integrations are negative or at best nearly-neutral with regards to the host. Only a tiny proportion of integrations are likely to be beneficial to the host in any way.

We completely agree with the reviewer and that's why we used "can" and not "are". However there are more and more evidences that some TE integrations have been beneficial to the host with numerous examples of cooption (Review on syncytins : (Dupressoir, Lavialle, and Heidmann 2012); VDJ review (Carmona and Schatz 2017); more general reviews:(Gasparotto et al. 2023; Nicolau, Picault, and Moissiard 2021; Cosby, Chang, and Feschotte 2019). We changed the phrasing of this part of the manuscript and added the reference (Nicolau, Picault, and Moissiard 2021): "ERVs are TEs and although the presence of TEs in the genome has been shown to provide some evolutionary benefits (reviewed in 7), unregulated TE expression and transposition represent a threat to genome integrity and host fitness." (line 55-58)

Line 28: when referring to the process of TE repression, the authors state that "Our results not only highlight how ERVs and their host adapt to each other"— how is the TE adapting to the host under this scenario?

In answer to this comment, we added a paragraph at the end of the Discussion explaining our hypothesis of how the ZAM errantivirus may have adapted to the host (lines 500-507).

Reviewer #2 (Remarks to the Author):

In this manuscript, Marianne et al., leveraged the RevI-H2 *Drosophila* line to study the reactivation of ZAM. The RevI-H2 derived isogenic line, RevI-H2i2, has a deletion (including ZAM sequence) in the major somatic piRNA cluster, resulting in ZAM derepression in follicle cells. Meanwhile, the RevI-H2i2 carries ZAM insertions into the germline piRNA clusters. Thus, RevI-H2i2 expresses germline piRNA to thwart ZAM invasion from follicle cells. By using CRISPR-Cas9, the ZAM specific deletion in *flam* results in ZAM invasion to oocytes and reduced female fertility. Since it is known that piRNAs are the major force to silence transposon in both somatic and germline cells of gonads, the selling point of this manuscript is the adaptive genomic immune response against the invading ERV. To strengthen it, the author should provide more evidence: 1. The follicle cells produced ZAM are capable of mobilizing into the genome of oocyte; 2. ZAM could insert into piRNA cluster, ultimately producing piRNA.

Major:

In Figure 1, although the ZAM transcripts accumulated about 3-fold in the follicle cells of RevI-H2i2 females compared to w¹¹¹⁸(Figure 1B, C), this strain is stable (Line 142-145, Figure 1E, F). Given RevI-H2i2 harbors both full-length and truncated copies of ZAM (Figure 1A), it is better to distinguish whether the accumulated ZAM transcripts are full-length or truncated by Nanopore RNA sequencing. Because only the full-length copies of ZAM are potentially mobile.

We thank the reviewer for this very relevant comment. The RevI-H2i2 line contains indeed full-length (named ZAM-fl) and truncated (named ZAM-v1, -v2 and -v3) copies of ZAM. 6 copies of the ZAM-fl and 4 copies of the ZAM-v2 and 2 copies of the ZAM-v3 are present in the genome of the RevI-H2i2 line strongly suggesting that at least ZAM-v2 and -v3 in addition to the ZAM-fl are competent for transposition. To test whether these copies are still expressed in the RevI-H2i2 genome we performed sequencing of the RevI-H2i2 ovary mRNAs. Interestingly, we observed that, besides the full-length ZAM, at least the ZAM variants ZAM-v2 and ZAM-v3 are expressed in RevI-H2i2 ovaries. These results are confirmed by RT-PCR experiments performed using specific primers of each variant to determine which copies are expressed. These new results are presented in the Supplementary Fig. 1c,d and have been reported in the text “lines 162-164)

Moreover, we have strong evidence that ZAM variants ZAM-fl, ZAM-v2 and ZAM-v3 are functional for transposition: adding three ZAM copies located on the X chromosome of the RevI-H2 line to the *flam*ΔZAM genome resulted in strong ZAM expression and oocyte invasion. Our new mRNA-Seq data show transcription of all three variants (see the new supplemental Figure 4b) Moreover, we have detected 125 new genome insertions including 43 ZAM-fl, 43 ZAM-v2 and 31 ZAM-v3 insertions demonstrating that these three ZAM variants are competent for transposition. Nevertheless, we prefer not to show these new insertions in this manuscript, for the following reasons: We don't yet know whether these transposition events occur in the germline, as we are finding them through whole-fly genomic sequencing, and we would like to start a new research project incorporating these data. We hope that the reviewers will understand our wish to preserve these results.

The RevI-H2i2 expresses germline piRNAs targeting ZAM (Figure 2A). Why there is about 10% of stage 10 cyst still have ZAM RNA in the ooplasm of white-GLKD (Figure 2E)?

As the Reviewer pointed out, 10% of stage 10 cysts contain indeed ZAM RNA in the oocyte in the RevIH2i2-white-GLKD condition suggesting that a minor invasion occurs in this condition. It's worth noting that for the purposes of counting the percentage of invasion, we considered any stage 10 follicles containing any ZAM signal to be positive for invasion, irrespective of the number or intensity of the signal. In the white-GLKD condition, only few spots were observed in the 10% of stage 10 follicles containing ZAM RNA, while knockdown of piRNA pathway actors resulted in a strong and widespread ZAM RNA signal throughout the ooplasm. Additionally, ZAM RNA was only found in the posterior pole of the ooplasm, close to the posterior follicle cells producing the ZAM transcripts, in the white-GLKD condition, suggesting that these RNAs may be degraded before being transported to the nucleus.

A new sentence has been added to the manuscript to point out this result:

“ZAM RNA staining was restricted to follicle cells, only a minor ZAM RNA signal was detected in the posterior pole of approximately 10% of stage 10 oocytes.” (lines 213-214)

It is a great idea to generate a single deletion of ZAM in flam (flamDZAM, Figure 4A-C). Strikingly, the flamDZAM carrying functional ZAM copies displayed atrophied ovaries, invaded ZAM transcripts, and 90% females are sterile (Figure 4E-F). Is this due to the transposition of ZAM in oocyte? If so, please quantify the new insertions.

As mentioned above, we searched for new ZAM insertions in a flam Δ ZAM line with 3 functional copies of ZAM on the X chromosome. We found 123 new insertions for a genome coverage of 121. This indicates that there is approximately one new ZAM insertion per genome. Still this is certainly underestimated since we used only reads >10 kb for our screen and retained only the reads that contained an entire ZAM element. Unfortunately, we don't know whether these ZAM insertions occur in the germline. We haven't mentioned these results in the manuscript, but if the reviewer thinks they are essential for the paper we can add them.

Unfortunately, it is impossible to construct a copy of ZAM with a retrotransposition indicator gene (Jensen S. and Heidmann T., 1991) because it is impossible to clone the entire ZAM element or even have its sequence synthesized. All our attempts have been in vain. This makes it very difficult to quantify new ZAM insertions in the germline. But we're going to set up a new project to do just that.

Although the RevI-H2i2 line happens to have the germline expressed piRNAs to prevent ZAM invasion, there is no detectable de novo piRNA being produced in flamDZAM (Figure 5A, B). How long does it take for the flies developing “adaptive genomic immune response against the constantly invading ERV”? It is better to keep raising the fertile flamDZAM-RevIH2i2 flies, evaluating their fertility and sequencing the ZAM piRNAs generation by generation.

The reviewer raised a crucial question here and we would indeed like to study in detail how long it takes to develop an adaptive genomic immune response against an invading TE. However, this study will take several months or even years to complete. We are currently in the process of studying different CRISPR flam Δ ZAM lines, which contain a variable number and different variants of ZAM copies by introducing different chromosomes of the RevI-H2i2 line (part of chromosome X, chromosome II or III). However, this leads to a significant reduction

in female fertility at least with the ZAM copies located on the X chromosome, which makes the inter-generational analysis more challenging. Nevertheless, we plan to examine fertility at each generation, the production of ZAM piRNA and even new genomic ZAM insertions. This comprehensive study will address numerous knowledge gaps related to the dynamics of invasion and transposition of invading TEs/errantiviruses.

Minor point

The *Drosophila* and gene name (*flam*, *w1118*...) should be in italics.

We thank the reviewer for pointing out this error. We have indeed corrected the gene names that should be in italics and were mistakenly left out.

Reviewer #3 (Remarks to the Author):

A class of gypsy retrotransposons has acquired an envelope gene to behave like a retrovirus in *Drosophila* species. It is known that these so-called errantivirus or envelope-carrying gypsy elements, including ZAM, are expressed in the ovarian somatic cells and infect oocytes. Thus, a distinct piRNA-based defence mechanism is in place in the ovarian somatic cells to suppress their expression. Authors previously showed that, a *Drosophila* strain that lacks the region that produces piRNAs against ZAM in the soma has acquired new insertions of ZAM elsewhere in the genome. They showed that those newly acquired insertions of ZAM produce piRNAs in the germline. In the current manuscript, Yoth et al demonstrated that those germline piRNAs target ZAM mRNAs that are expressed in the soma and transmitted to the germline.

Several studies have examined piRNA-based adaptation processes in *Drosophila* after an acquisition of new transposons. But the current study is unique in a sense that the niche where transposons are activated is different from the cells where the silencing occurs. Therefore, it adds a significant value to the existing body of literature. I recommend a publication in Nature Communications should the following comments be adequately addressed.

1. There is evidence that somatic gypsy elements turn into germline transposons over time. Can you rule out that some of new ZAM insertions in the RevI-H2 are now expressed in the germline, and those are the ones that are targeted by the germline-expressed piRNAs? Along this line, there are weak but significant nuclear FISH signals of ZAM mRNA in the nurse cells even in the control knockdown in Figure 2. Are they reproducible? Are they not seen in the control strain?

This is a very interesting point, which we did not discuss in the paper to avoid confusion but the potential expression of ZAM in the germline is a major point that we have given much thought. There is strong evidence that ZAM has indeed not acquired the ability to be expressed in germ cells of the RevI-H2i2 line.

First, we did not detect any modifications in the promoter of the various ZAM genomic copies present in the RevI-H2i2 genome that would suggest acquisition of new regulatory sequences enabling its expression in germline cells. Since ZAM expression is controlled by the transcription factor Pointed that is only expressed in follicular cells, it is unlikely that ZAM start to be expressed in the germ cells.

We did detect a ZAM RNA signal in the nurse cells' nuclei in the RevI-H2i2 line in all observed follicles, but we have strong evidence suggesting that these dots are the result of the expression of the germline piRNA clusters containing a ZAM insertion in the RevI-H2i2 line:

- Firstly, these nuclear foci are not detected in any other genetic background (w^{1118} , wIR6, white-sKD, vret-sKD...) where ZAM insertions in a germline piRNA cluster are not present.
- The foci are also detected when using an smFISH probe targeting antisense ZAM transcripts, which should be detected if we observe the expression of a dual strand piRNA cluster (data not shown). We feel that this information can be confusing when reading the manuscript, but we can add in the manuscript the following information if the reviewer thinks it adds value to the manuscript:

“We noted some ZAM FISH signal in the nucleus using probes targeting both sense and antisense ZAM RNA probably reflecting the expression of the dual-strand germline piRNA cluster containing ZAM insertion in the RevI-H2i2 line”.

Interestingly, nuclear foci have also been observed for the I-element (in inducer females), and I-element RNA appears to be trapped in the nuclei due to the production of I-element piRNAs in germ cells (Chambeyron et al 2008). In the absence of I-element piRNAs (RSF females or GLKD of the piRNA pathway), I-element transcripts are detected in the cytoplasm of nurse cells and in the ooplasm.

In the RevI-H2 line, the GLKD of piRNA pathway actors does not affect the ZAM RNA pattern in nurse cells. ZAM RNAs are still only detected in the nuclei of nurse cells and in the ooplasm; no ZAM RNA is detected in the cytoplasm of nurse cells.

Finally, one important point raised by the reviewer in the next comment is that despite the requirement of Piwi for the transcriptional silencing of TEs, we did not observe any derepression of ZAM following Piwi-GLKD in the RevI-H2i2 condition. This observation suggests that germline ZAM piRNAs are not involved in the transcriptional silencing of ZAM but rather in the post-transcriptional silencing of ZAM mRNA that originates from somatic cells.

We have addressed this point by adding the paragraph at lines 244-258 and adding figure 3.

2. Authors showed that a germline knockdown of Piwi does not derepress ZAM in the RevI-H2 background (Figure 2). I find this data point crucial because it supports the idea that the germline piRNAs target ZAM mRNAs from the soma. However, I do not see the validation of this RNAi line. Can you show that other germline transposons are derepressed in the same cross?

We have decided to emphasize this result that indeed supports the idea of post-transcriptional silencing of ZAM mRNA arriving from the soma by germline piRNAs. We added a new paragraph in the manuscript that now includes the results of RevI-H2i2 Piwi-GLKD, as well as controls to show that other germline transposons are derepressed in this condition, while ZAM RNA level and localization remain unaffected. smFISH staining of Burdock RNA and RT-qPCR clearly showed a significant derepression of this TE following Piwi-GLKD, validating the RNAi line (new Figure 3).

3. Please outline individual egg chambers in FISH images when the borders are not clear. Images contain information of where in each egg chamber ZAM is expressed. It is hard to work it out without seeing the structure of individual egg chambers. This applies, for example, to Figure 1B.

We agree that the individual egg chambers were hardly identifiable in some figures. We have therefore changed the presentation of some figures by adding a DAPI staining to visualize the nuclei of the different cells in each egg chamber (Fig 1b and 2g).

4. The antibody staining of ZAM proteins in Figure 1D is not very clear. Please include a staining of control egg chamber that does not express ZAM to demonstrate specificities.

Control egg chamber of the w^{1118} line have been added to Figure 1d.

5. Likewise for Figure S2A, please include a control ovariole where the pGFP-ZAM reporter is suppressed in the soma.

Control ovariole of the w^{1118} line have been added to Figure S2A.

6. Please make available the code used for the computational analyses. For example, it is not indicated how the RPKM was calculated for the data presented in Figure 2B. For this reason, I cannot evaluate the differences between different TEs. In fact, it appears that the piRNAs against the somatic transposon Gypsy10 have decreased upon knockdown of ago3 or zucchini in the germline.

We used the following calculation for RPM :

$$\text{RPM} = \text{read-count} * 1,000,000 / (\text{total of genome mapping piRNAs})$$

We have also included this information in the Material and Methods section (lines 641-643).

To deal with the potential decrease of piRNA targeting somatic TEs upon knockdown of ago3 and zucchini we examined other somatic TEs, such as Tirant or Tabor.

We observed that Ago3 GLKD did not result in a widespread decrease of piRNAs targeting somatic TEs. The reduction of Gypsy10 piRNAs is negligible, and piRNAs targeting Tirant or Tabor are unaffected by Ago3-GLKD. Consequently, we decided to present Tirant as a somatic TE in the Fig 2b instead of Gypsy10.

However, we encountered an issue with the Zuc-GLKD condition, as we observed abnormal results when closely examining the small RNA sequencing data. In particular, there was a general decrease in piRNAs targeting all TEs, even somatic TEs. In light of this, we decided to repeat the small RNA sequencing for the RevI-H2i2 Zuc-GLKD ovaries, employing the TRAPR method to specifically isolate regulatory small RNAs. We found that piRNAs targeting somatic TEs were not affected by Zuc-GLKD, while a significant decrease was observed for germline TEs and ZAM piRNAs.

Therefore, we are now presenting the results from RevI-H2i2 Zuc-GLKD regulatory piRNAs in a separate supplementary Fig. 2b, as they cannot be directly compared to the results obtained from the previous total RNA extractions presented in Fig.2b as the extraction method is different.

7. It is helpful to indicate which ZAM insertions are thought to be active and which are producing piRNAs in the germline. I appreciate that one cannot nail down to individual insertions. However, the authors' previous publication showed that the X chromosome is responsible for making the germline piRNAs. This information should be clearly noted in this current manuscript. Otherwise, the experiments using flam-delta-ZAM + RevI-H2 chromosomes cannot be interpreted.

For the same reason, please describe all the crosses that partially include RevI-H2i2 chromosomes to the chromosome level. For example, the RNAi lines used in the study are on different chromosomes. The testing flies may contain different combinations of RevI-H2i2-derived chromosomes depending on the crossing scheme.

We thank the reviewer for this very relevant comment. Indeed, we intended to simplify the information for the reader, but we fully agree that the information about the origin of ZAM piRNA production in the RevI-H2 line is essential for a complete understanding of the experiments and results. We are also aware that genetic background can introduce many biases into the analysis, and we have taken great care to control for this aspect as best as possible.

However, it is true that the number of ZAM insertions varies among the tested flies, and it is important to highlight this in the manuscript.

Therefore, we have added the following information in the manuscript:

Line 187: Importantly, our previous research has demonstrated that the X chromosome of the RevI-H2 line, which contains the ZAM copies inserted in germline piRNA clusters, is both necessary and sufficient to produce ZAM-regulating piRNAs²⁸.

Lines 200-203: It is important to note that, during the genetic crosses performed, only the X chromosome of the RevI-H2i2 was tracked to ensure that the ZAM insertions in the germline piRNA clusters were maintained (crossing schemes presented in Figure S6).

Line 318-320: We made sure not to introduce the ZAM copies located in germline piRNA clusters of the X chromosome of the RevI-H2i2 line that are involved in the production of ZAM-regulating piRNAs in the germline.

Line 325 : when we added three euchromatic ZAM copies ...

Additionally, we have added Figure S6, which displays all the genetic crosses that partially include RevI-H2i2 chromosomes such as those used to generate the RevI-H2i2 Nanos-Gal4 and the RevI-H2i2 RNAi lines. This will help to better understand the diversity in ZAM copies that may exist in the tested flies.

Additional minor changes made to the manuscript:

- One new ZAM variant, ZAM-v3, has been found in the RevI-H2i2 genome. Therefore, we have now included this variant to the schematic representations of all variants and have indicate the genomic insertions of this variant in figure 1a. In this new version of the manuscript, we present the expression of the different ZAM variants, so we thought it would be helpful to include the schematic representation of the ZAM variants in the main figure 1a.
- Genomic sequencing of the $\text{flam}\Delta\text{ZAM}$ line, which carries ZAM copies from the X chromosome of RevI-H2, has revealed that we actually introduced three copies of ZAM instead of two, as initially stated in the first manuscript. Therefore, we have made all the necessary corrections and adjustments based on this new information.
- We have moderated our statements concerning the sterility observed after ZAM reactivation, as sterility is not observed with all ZAM copies introduced on different chromosomes.
- In Supplementary Table S1, the variants ZAM-v1 and ZAM-v2 were indicated and their attributes updated from recent complementary results.
- In Supplementary Table S2, we added a title and corrected minor spelling errors.

- In Supplementary Table S5, we added the primers that were used for the detection of transcripts of the different ZAM variants (Supplementary figure 4b).

REFERENCES :

- Boeke, J. D., and J. P. Stoye. 1997. "Retrotransposons, Endogenous Retroviruses, and the Evolution of Retroelements." In , edited by John M. Coffin, Stephen H. Hughes, and Harold E. Varmus.
- Capy, P. 2005. "Classification and Nomenclature of Retrotransposable Elements." *Cytogenetic and Genome Research* 110 (1–4): 457–61.
- Carmona, Lina Marcela, and David G. Schatz. 2017. "New Insights into the Evolutionary Origins of the Recombination-Activating Gene Proteins and V(D)J Recombination." *The FEBS Journal* 284 (11): 1590–1605.
- Chambeyron S, Popkova A, Payen-Groschêne G, Brun C, Laouini D, Pelisson A, and Bucheton A. 2008 " piRNA-mediated nuclear accumulation of retrotransposon transcripts in the Drosophila female germline" *Proc Natl Acad Sci.* 105(39) : 14964–14969.
- Cosby, Rachel L., Ni-Chen Chang, and Cédric Feschotte. 2019. "Host-Transposon Interactions: Conflict, Cooperation, and Cooption." *Genes & Development* 33 (17–18): 1098–1116.
- Dupressoir, A., C. Laviolle, and T. Heidmann. 2012. "From Ancestral Infectious Retroviruses to Bona Fide Cellular Genes: Role of the Captured Syncytins in Placentation." *Placenta* 33 (9): 663–71.
- Gasparotto, Erica, Filippo Vittorio Burattin, Valeria Di Gioia, Michele Panepuccia, Valeria Ranzani, Federica Marasca, and Beatrice Bodega. 2023. "Transposable Elements Co-Option in Genome Evolution and Gene Regulation." *International Journal of Molecular Sciences* 24 (3). <https://doi.org/10.3390/ijms24032610>.
- Guedán, Anabel, Eve R. Caroe, Genevieve C. R. Barr, and Kate N. Bishop. 2021. "The Role of Capsid in HIV-1 Nuclear Entry." *Viruses* 13 (8). <https://doi.org/10.3390/v13081425>.
- Jensen S. and Heidmann T., 1991, "An indicator gene for detection of germline retrotransposition in transgenic Drosophila demonstrates RNA-mediated transposition of the LINE I element" *EMBO Journal* 10(7)
- Llorens, Carlos, Beatriz Soriano, Mart Krupovic, and Ictv Report Consortium. 2020. "ICTV Virus Taxonomy Profile: Metaviridae." *The Journal of General Virology* 101 (11): 1131–32.
- Malik, H. S., S. Henikoff, and T. H. Eickbush. 2000. "Poised for Contagion: Evolutionary Origins of the Infectious Abilities of Invertebrate Retroviruses." *Genome Research* 10 (9): 1307–18.
- Nicolau, Melody, Nathalie Picault, and Guillaume Moissiard. 2021. "The Evolutionary Volte-Face of Transposable Elements: From Harmful Jumping Genes to Major Drivers of Genetic Innovation." *Cells* 10 (11). <https://doi.org/10.3390/cells10112952>.
- Stefanov, Yury, Veniamin Salenko, and Ivan Glukhov. 2012. "Drosophila Errantiviruses." *Mobile Genetic Elements* 2 (1): 36–45.

Syomin, B. V., K. V. Kandror, A. B. Semakin, V. L. Tsuprun, and A. S. Stepanov. 1993. "Presence of the Gypsy (MDG4) Retrotransposon in Extracellular Virus-like Particles." *FEBS Letters* 323 (3): 285–88.

Wicker, Thomas, François Sabot, Aurélie Hua-Van, Jeffrey L. Bennetzen, Pierre Capy, Boulos Chalhoub, Andrew Flavell, et al. 2007. "A Unified Classification System for Eukaryotic Transposable Elements." *Nature Reviews. Genetics* 8 (12): 973–82.

Reviewers' Comments:

Reviewer #1:

Remarks to the Author:

I thank the authors for altering the retroviral focus of their manuscript, I believe the changes will help to avoid propagating confusion in the field. I have a few other specific suggestions below, which I believe will further improve clarity for the reader. I hope the authors will find these useful. I apologise if these appear tedious, but the biology of this system is complex, and it is my hope that clarifying the aspects below will help to clear up several key issues and make the manuscript accessible to a general audience.

Clarification of errantivirus transfer between somatic and germ cells (introduction): Given the focus of this study, I think it is important that the authors discuss what is known of the transfer of errantiviruses between somatic and germline cells in the introduction. Specifically, I suggest clarifying early on the distinction between the infectious viral pathway conferred by the envelope-like gene, and the situation where env is lacking but virus-like particles are produced, outlining the potential routes by which ZAM may enter the germline and their connotations. At present, the new paragraph discussing the focus of the study (L92-96) goes beyond the presented background. Meanwhile, throughout the manuscript, there is a focus on ZAM RNAs, and there should be clarification, so that the reader understands key aspects of ZAM biology from the onset, such as that various proteins encoded in the ZAM genome are required for integration into the host genome. For example, phrasing relating to the transit of ZAM RNAs is somewhat misleading, as it is not naked RNA that is transiting between cells. The current discussion of these aspects comes too late in the manuscript (discussion, L447-477), and could be organised more coherently in a section in the introduction.

Terminology: The authors have made a strong effort to rephrase the retrovirus focussed language of the previous manuscript, but there are still some issues with terminology. I suggest the authors do not refer to ERVs as a 'family' (L33). ERVs are massively diverse, and would consist of many families if applying the term in the usual TE sense. The authors use family in a more in keeping context elsewhere, when stating that each ERV copy can result in the emergence of a new TE family (L43-44). Immediately after the implication that ERVs are a family, the authors go on to state that ERVs are a 'class' of TEs (L34), that are closely related to retroviruses. What is the usage of class supposed to convey and what useful information do we get by using this term? Meanwhile, it is somewhat misleading to state that ERVs are 'closely related to infectious retroviruses', as this could be interpreted to mean that they are evolutionarily distinct from retroviruses. I suggest a more useful/appropriate framework, both for the reader and the study, would be to highlight the biology, specifically that ERVs derive directly from retroviruses that insert into the host germline and become fixed in the host lineage, and are thus a partial record of retroviral infection history and evolution.

Structure of the introduction: Rather than starting the manuscript with a paragraph on retrovirus biology, it would likely probably be more useful and appropriate to start by summarising errantivirus biology. Also, perhaps Pélisson et al (2002) ([https://doi.org/10.1016/S0965-1748\(02\)00088-7](https://doi.org/10.1016/S0965-1748(02)00088-7)) should be cited somewhere in the introduction, since although focussed on gypsy, the discussions are highly relevant?

Host-TE adaptation framework: The authors state "Our results not only highlight how errantiviruses and their host adapt to each other" (L27) and they offer further discussion of this topic on L500-L507. This framework and how the specific results highlight would benefit from more explanation/discussion. The deletion of ZAM in flamenco is not a good example of adaptation of a TE to a host, as it rests upon the failure of a host defence mechanism, which is independent of ZAM. Tissue specific expression restricted to somatic gonal cells could be used as an example of adaptation, but what of the severe effects on fertility of the host and their connotations for the fitness of ZAM? Meanwhile, I am not entirely clear on the language the authors use when they state that the germline "set up its own adaptive genomic immune response" after ZAM reactivation, as the deployment of an existing

germline piRNA response (even to ZAM generated in the soma) would not be considered adaptation. Additionally, the ending of the final paragraph in the manuscript suggests an evolutionary victory for ZAM, while evidence suggests that such elements are typically efficiently shut down and unable to evade/adapt to host defences.

Minor suggestions:

L18 (and elsewhere): 'Errantivirus' should be italicized.

L19: It is unclear what 'These' refers to, i.e. which virus-derived elements - Errantiviruses in particular? It is also unclear which 'genome' - the genomes of species in *Drosophila*? For example, *D. simulans* and *D. virilis* contain something like 5% Gypsy elements, is this a large part of the genome? Also, what is meant by 'it is unclear whether and how they can be reactivated and if they retain their replication capacity'? This statement is confusing regarding different potential modes of replication, on the one hand the intracellular replication of transposition and on the other the infectious viral lifecycle. I suggest the authors are more specific about their precise meaning here.

L22: what is meant by 'in real time' exactly?

L26: "The germline then set up its own adaptive genomic immune response" - I suggest expanding on this statement to state precisely what happened.

L39-41: It might be worth clarifying that it is generally thought that most ERVs are not able to produce competent virions for long.

L50: what is meant "from time to time"?

L52: "TEs that have acquired the capacity to be expressed exclusively in somatic cells must therefore maintain their ability to invade germ cells to survive." unclear.

L57: what are "physiological conditions"?

L58: "maintained in a dormant state" - I am not sure of the use of dormancy to explain repressed TEs here, this phrase could just be deleted.

L69: 'Metaviridae' should be italicized.

L98: What is the meaning of "still" here? Also, "doesn't" to 'does not'.

L107: "the control" to 'control'?

L108: What are "long periods of time" here?

L194: "TEs, such as ZAM, are not expressed directly in germ cells but arrive from surrounding somatic cells" - is this indeed a widespread strategy among TEs?

L208: "while somatic-TEs such as Tirant did not show a similar decrease" unclear.

L238: "This result confirmed that, when no ZAM piRNA is produced in the germline, ZAM RNAs expressed in the somatic follicle cells transit to the oocyte." Does it not confirm that the RNAs transit and also persist?

Reviewer #2:

Remarks to the Author:

The authors addressed my concerns. While some of the supporting data were not included in this version, I understand the authors have chosen to preserve it for future publication. I would suggest the authors pay attention to the minor formatting issue: the term "Drosophila" should be italicized.

Reviewer #3:

Remarks to the Author:

I appreciated the changes made in the revised manuscript and would like to recommend a publication in principle with a couple of minor suggestions.

1. The aesthetics of the new Figure 3 looks odd in the pdf. Please correct it. The scale bars in the new Figure 3a are invisible. Please correct it.

2. I appreciate that the staining of the GFP-ZAM reporter in the control strain is now added in the Supplementary Figure 2a. The striking difference between the control and the Rev-H2i2 is not explained (although it biologically makes sense) anywhere in the manuscript. Please add a brief explanation in the figure legend.

3. "The foci are also detected when using an smFISH probe targeting antisense ZAM transcripts, which should be detected if we observe the expression of a dual strand piRNA cluster (data not shown). We feel that this information can be confusing when reading the manuscript, but we can add in the manuscript the following information if the reviewer thinks it adds value to the manuscript:"

I agree with the authors. Please do not include the additional information in the manuscript.

Reviewer 1

I thank the authors for altering the retroviral focus of their manuscript, I believe the changes will help to avoid propagating confusion in the field. I have a few other specific suggestions below, which I believe will further improve clarity for the reader. I hope the authors will find these useful. I apologise if these appear tedious, but the biology of this system is complex, and it is my hope that clarifying the aspects below will help to clear up several key issues and make the manuscript accessible to a general audience.

We would like to thank the reviewer for the quality of his proofreading, the time he devoted to it and his invaluable help, which enabled us to improve our manuscript by being more precise.

Clarification of errantivirus transfer between somatic and germ cells (introduction): Given the focus of this study, I think it is important that the authors discuss what is known of the transfer of errantiviruses between somatic and germline cells in the introduction. Specifically, I suggest clarifying early on the distinction between the infectious viral pathway conferred by the envelope-like gene, and the situation where env is lacking but virus-like particles are produced, outlining the potential routes by which ZAM may enter the germline and their connotations. At present, the new paragraph discussing the focus of the study (L92-96) goes beyond the presented background. Meanwhile, throughout the manuscript, there is a focus on ZAM RNAs, and there should be clarification, so that the reader understands key aspects of ZAM biology from the onset, such as that various proteins encoded in the ZAM genome are required for integration into the host genome. For example, phrasing relating to the transit of ZAM RNAs is somewhat misleading, as it is not naked RNA that is transiting between cells. The current discussion of these aspects comes too late in the manuscript (discussion, L447-477), and could be organised more coherently in a section in the introduction.

As suggested by the reviewer, we have now included in the introduction an explanation of what is currently known about the transit of errantiviruses from somatic cells to germ cells. We recognize that while we do not directly address the mechanism employed by errantivirus for germline invasion in this study, this information is important for a comprehensive understanding of the manuscript.

Apart from ZAM and Gypsy, there is limited information available on this topic. We emphasize that the precise role of Env in oocyte infection remains unclear and explain that an Env-independent pathway might also be involved. Additionally, we mention that most errantiviruses contain Gag, Pol and Env ORF and that viral-like particles have been observed in follicle cells and likely contain the RNA of these elements. With this new information and the sentence at line 194 stating that "ZAM RNA may transit in an encapsulated form," it will likely assist the reader in better comprehending that the transit probably involves RNA being encapsulated.

The following information has been added in the introduction:

L 38 - "Most of them encode 3 ORFs whose functions are analogous to that of the Gag, Pol and Env proteins of ERVs or retroviruses."

L 63-68 - "Gypsy and ZAM virus-like particles have been observed in somatic follicle cells. They likely contain the genomic RNA and the expression of the envelope protein could mediate an infection process^{8,17}. However, the Gypsy envelope is not involved in the soma-to-germline transfer¹⁵. Instead, it has been proposed that not only Gypsy but also ZAM hijack the host vitellogenic pathway to target the oocyte^{11,18}."

Terminology: The authors have made a strong effort to rephrase the retrovirus focussed language of the previous manuscript, but there are still some issues with terminology. I suggest the authors do not refer to ERVs as a 'family' (L33). ERVs are massively diverse, and would consist of many families if applying the term in the usual TE sense. The authors use family in a more in keeping context elsewhere, when stating that each ERV copy can result in the emergence of a new TE family (L43-44). Immediately after the implication that ERVs are a family, the authors go on to state that ERVs are a 'class' of TEs (L34), that are closely related to retroviruses. What is the usage of class supposed to convey and what useful information do we get by using this term? Meanwhile, it is somewhat misleading to state that ERVs are 'closely related to infectious retroviruses', as this could be interpreted to mean that they are evolutionarily distinct from retroviruses. I suggest a more useful/appropriate framework, both for the reader and the study, would be to highlight the biology, specifically that ERVs derive directly from retroviruses that insert into the host germline and become fixed in the host lineage, and are thus a partial record of retroviral infection history and evolution.

As suggested by the reviewer we do not refer anymore to ERVs as a “family” or as ‘closely related to infectious retroviruses’ and we have restructured this part of the introduction to enhance its comprehensibility:
L 40-41 “ERVs derive from exogenous retroviruses that integrated into the host germline genome, became permanent elements and are now vertically transmitted.”

We also consider the reviewer's statement regarding ERVs as a partial record of retroviral infection to be highly relevant. Therefore, we included this information in the introduction.
L 41-42 : “Therefore, they represent a partial record of previous retroviral infections”

Structure of the introduction: Rather than starting the manuscript with a paragraph on retrovirus biology, it would likely probably be more useful and appropriate to start by summarising errantivirus biology. Also, perhaps Pélisson et al (2002) ([https://doi.org/10.1016/S0965-1748\(02\)00088-7](https://doi.org/10.1016/S0965-1748(02)00088-7)) should be cited somewhere in the introduction, since although focussed on gypsy, the discussions are highly relevant?

We agree that it makes more sense to begin by introducing errantivirus, as we now clearly distinguish between ERVs and errantiviruses. Therefore, we have modified the introduction to start by defining errantiviruses before discussing ERVs (L34-42). We then highlight similarities and differences between ERV and errantivirus biology. We have also added the citation “Pelisson et al, 2002”, which is, we agree, highly relevant.

Host-TE adaptation framework: The authors state “Our results not only highlight how errantiviruses and their host adapt to each other” (L27) and they offer further discussion of this topic on L500-L507. This framework and how the specific results highlight would benefit from more explanation/discussion. The deletion of ZAM in flamenco is not a good example of adaptation of a TE to a host, as it rests upon the failure of a host defence mechanism, which is independent of ZAM. Tissue specific expression restricted to somatic gonadal cells could be used as an example of adaptation, but what of the severe effects on fertility of the host and their connotations for the fitness of ZAM? Meanwhile, I am not entirely clear on the language the authors use when they state that the germline “set up its own adaptive genomic immune response” after ZAM reactivation, as the deployment of an existing germline piRNA response (even to ZAM generated in the soma) would not be considered adaptation. Additionally, the ending of the final paragraph in the manuscript suggests an evolutionary victory for ZAM, while evidence suggests that such elements are typically efficiently shut down and unable to evade/adapt to host defences.

Following the reviewer's comments and also because of space constraints, we can't afford to discuss this part with more explanation and detail, and it's probably not the subject of our article. We therefore prefer to remove from the discussion the last part, which covers the co-evolution of host and TE.

The consequence of this change is also reflected in the abstract, where we no longer refer to “how errantiviruses and their host adapt to each other but also reveal a time window during oogenesis that may be favourable for viral germline invasion and endogenization”. We therefore remain more pragmatic in the conclusion of our abstract, focusing on the results obtained. “Our results underline that errantiviruses, although constantly repressed by the piRNA pathway, may retain their ability to infect the germline and transpose, allowing them to efficiently invade the germline if they are misguidedly expressed.”

Minor suggestions:

L18 (and elsewhere): ‘Errantivirus’ should be italicized.
This change has been done.

L19: It is unclear what ‘These’ refers to, i.e. which virus-derived elements - Errantiviruses in particular? It is also unclear which ‘genome’ - the genomes of species in *Drosophila*? For example, *D. simulans* and *D. virilis* contain something like 5% Gypsy elements, is this a large part of the genome? Also, what is meant by ‘it is unclear whether and how they can be reactivated and if they retain their replication capacity’? This statement is confusing regarding different potential modes of replication, on the one hand the intracellular replication of transposition and on the other the infectious viral lifecycle. I suggest the authors are more specific about their precise meaning here.

We agree that various information in the abstract were confusing and not precise and decided to make the following modifications according to the reviewer advice. We wrote a shorter but more precise sentence “Like ERVs, it is unclear whether errantiviruses retain some infectivity and transposition capacity. “
Due to word count constraints in the summary, we can't give any further details.

L22: what is meant by ‘in real time’ exactly?
We have changed “reactivation in real time” to “*de novo* reactivation”

L26: “The germline then set up its own adaptive genomic immune response” - I suggest expanding on this statement to state precisely what happened.

We have added “by producing piRNAs” to precise our statement. Due to word count constraints in the summary, we can't give any further details.

L39-41: It might be worth clarifying that it is generally thought that most ERVs are not able to produce competent virions for long.

We have added this sentence L 50-51: “Notably, the majority of ERVs are incapable of producing infectious viral particles *in vivo*.”

L50: what is meant “from time to time”?

We agree with the reviewer that the term “from time to time” is not precise, so we have removed it.

L52: “TEs that have acquired the capacity to be expressed exclusively in somatic cells must therefore maintain their ability to invade germ cells to survive.” unclear.

We have reorganized this part of the introduction to clarify our statement and now give the example of ZAM, Idefix and Gypsy that are only expressed in somatic cells:

L 57-63 : “For instance, in the *Drosophila melanogaster* genome, ZAM, Idefix or Gypsy errantiviruses are expressed exclusively in the somatic follicle cells of the ovaries, their expression being controlled by somatic transcription factors⁹⁻¹⁵. The *Drosophila melanogaster* ovary comprises about 16 ovarioles, each of which contains a succession of follicles composed of germ cells surrounded by somatic follicle cells¹⁶. To reach the germline genome, these errantiviruses must then cross the so-called Weismann barrier separating somatic and germ cells.”

L57: what are “physiological conditions”? L58: “maintained in a dormant state” - I am not sure of the use of dormancy to explain repressed TEs here, this phrase could just be deleted.

As suggested by the reviewer the entire sentence has been removed.

L69: ‘Metaviridae’ should be italicised.

The correction has been done.

L98: What is the meaning of “still” here? Also, “doesn’t” to ‘does not’.

We replaced “still” by “ZAM expression persists” and the correction to “does not” has been made.

L107: “the control” to ‘control’?

The correction has been made

L108: What are “long periods of time” here?

We have clarified our statement and added “(in the case of the Rev1-H2 line at least 30 years)”.

L194: “TEs, such as ZAM, are not expressed directly in germ cells but arrive from surrounding somatic cells” - is this indeed a widespread strategy among TEs?

We acknowledge the reviewer's comment that the strategy described is not widespread among TEs, but rather limited to some TEs. We apologize for any confusion and have made the necessary clarification by adding "some" before "TEs".

L208: “while somatic-TEs such as Tirant did not show a similar decrease” unclear.

To clarify that point the sentence has been rephrased: “The decrease in antisense piRNAs was comparable to that observed for germline TEs (i.e., Burdock and Accord) or intermediate TEs (i.e., Idefix), while the level of piRNA of somatic TEs such as *Tirant* was not affected.”

L238: “This result confirmed that, when no ZAM piRNA is produced in the germline, ZAM RNAs expressed in the somatic follicle cells transit to the oocyte.” Does it not confirm that the RNAs transit and also persist?

We thank the reviewer for this helpful comment. We have rephrased the sentence as follows: “This result confirmed that, when no ZAM piRNA is produced in the germline, ZAM RNAs expressed in the somatic follicle cells transit to the oocyte and then persist there”

Reviewer #2 (Remarks to the Author):

The authors addressed my concerns. While some of the supporting data were not included in this version, I understand the authors have chosen to preserve it for future publication. I would suggest the authors pay attention to the minor formatting issue: the term “*Drosophila*” should be italicized.

We would like to thank the reviewer for this second review round.

Drosophila has been italicized.

Reviewer #3 (Remarks to the Author):

I appreciated the changes made in the revised manuscript and would like to recommend a publication in principle with a couple of minor suggestions.

We would like to thank the reviewer for this second review round.

1. The aesthetics of the new Figure 3 looks odd in the pdf. Please correct it. The scale bars in the new Figure 3a are invisible. Please correct it.

We have done the correction.

2. I appreciate that the staining of the GFP-ZAM reporter in the control strain is now added in the Supplementary Figure 2a. The striking difference between the control and the Rev-H2i2 is not explained (although it biologically makes sense) anywhere in the manuscript. Please add a brief explanation in the figure legend.

We have added a brief explanation in the legend of the Supplementary Figure 2a: "In *w¹¹¹⁸* control ovaries, the transgene was completely silenced in somatic cells and strongly expressed in germline cells. Conversely, in Revl-H2i2 ovaries, the transgene was silenced in the germline and strongly expressed in somatic cells indicating that ZAM-derived piRNAs produced in the Revl-H2i2 germline cells efficiently guide sensor silencing in these cells."

3. "The foci are also detected when using an smFISH probe targeting antisense ZAM transcripts, which should be detected if we observe the expression of a dual strand piRNA cluster (data not shown). We feel that this information can be confusing when reading the manuscript, but we can add in the manuscript the following information if the reviewer thinks it adds value to the manuscript:"

I agree with the authors. Please do not include the additional information in the manuscript.

No changes requested by the reviewer.

Reviewers' Comments:

Reviewer #1:

Remarks to the Author:

I am satisfied with the changes made to the manuscript by the authors.